*GENETICS*, 2024, **226(4)**, iyae024

**Investigation**

# Interpreting generative adversarial networks to infer natural selection from genetic data

Rebecca Riley,[1] Iain Mathieson [ID],[2] Sara Mathieson [ID] [1,*]

[1]Department of Computer Science, Haverford College, Haverford, PA 19041, USA
[2]Department of Genetics, Perelman School of Medicine, University of Pennsylvania, Philadelphia, PA 19104, USA

*Corresponding author: Department of Computer Science, Haverford College, Haverford, PA 19041, USA. Email: smathieson@haverford.edu

Understanding natural selection and other forms of non-neutrality is a major focus for the use of machine learning in population genetics. Existing methods rely on computationally intensive simulated training data. Unlike efficient neutral coalescent simulations for demographic inference, realistic simulations of selection typically require slow forward simulations. Because there are many possible modes of selection, a high dimensional parameter space must be explored, with no guarantee that the simulated models are close to the real processes. Finally, it is difficult to interpret trained neural networks, leading to a lack of understanding about what features contribute to classification. Here we develop a new approach to detect selection and other local evolutionary processes that requires relatively few selection simulations during training. We build upon a generative adversarial network trained to simulate realistic neutral data. This consists of a generator (fitted demographic model), and a discriminator (convolutional neural network) that predicts whether a genomic region is *real* or *fake*. As the generator can only generate data under neutral demographic processes, regions of real data that the discriminator recognizes as having a high probability of being "real" do not fit the neutral demographic model and are therefore candidates for targets of selection. To incentivize identification of a specific mode of selection, we *fine-tune* the discriminator with a small number of custom non-neutral simulations. We show that this approach has high power to detect various forms of selection in simulations, and that it finds regions under positive selection identified by state-of-the-art population genetic methods in three human populations. Finally, we show how to interpret the trained networks by clustering hidden units of the discriminator based on their correlation patterns with known summary statistics.

Keywords: population genetics; machine learning; generative adversarial networks; natural selection; interpretability

## Introduction

In the last few years, deep learning methods have been proposed as solutions to numerous population genetic problems (see Korfmann *et al.* 2023 for a recent review), in part due to the ability of deep neural network architectures to recover evolutionary signals from noisy population-level data. Convolutional neural networks (CNNs) have been particularly effective and allowed the field to move away from summary statistics as inputs and toward analyzing genetic data directly. The broad idea behind CNNs for genetic data is to treat genotype matrices (with individuals or haplotypes in the rows and sites/SNPs in the columns) as analogous to images, with convolutional filters picking up on correlations between nearby sites. Population genetic CNNs have been developed for a variety of applications, including recombination hotspot identification (Chan *et al.* 2018), introgression, recombination rates, selection, and demography (Flagel *et al.* 2018), natural selection (Torada *et al.* 2019; Qin *et al.* 2022), adaptive admixture and introgression (Gower *et al.* 2021; Hamid *et al.* 2023), balancing selection (Isildak *et al.* 2021), dispersal inference (Smith *et al.* 2023), and the task-agnostic software dnadna (Sanchez *et al.* 2023). New architectures (including those built upon CNNs) have also been proposed, including graph neural networks (Korfmann *et al.* 2022), U-Nets (Ray *et al.* 2023), and recurrent neural networks (Adrion, Galloway, *et al.* 2020; Hejase *et al.* 2022).

One drawback of these machine learning approaches is that they require simulated training data, since labeled examples (i.e. where the historical evolutionary "ground truth" is known) are limited. If the simulated training data are not a good match for the real genetic data, inference results may not be robust. This is a particular problem for understudied populations and non-model species, where less is known about their evolutionary parameter space. Several solutions to this "simulation mis-specification" problem have been proposed, including adaptive re-weighting of training examples (Burger *et al.* 2022) and domain-adaptive neural networks (Mo and Siepel 2023). Another strategy is to create custom simulations using generative adversarial networks (GANs) (Goodfellow *et al.* 2020). Our previous work (Wang *et al.* 2021) described a GAN (`pg-gan`) that fits an evolutionary model to any population—as training progresses, the model produces synthetic data that are closer and closer to the real data. The key innovation of `pg-gan` is that it learns an explicit evolutionary generative model, in contrast to other GANs which generate sequences that look like real data from random processes with no underlying model (Yelmen *et al.* 2021; Booker *et al.* 2023). Recent work has made use of `pg-gan` for other species (Small *et al.* 2023) and improved the approach using adversarial Monte Carlo methods (Gower *et al.* 2023).

GANs have two components, a generator which creates synthetic data, and a discriminator which is fed examples of both

real and synthetic ("fake") data and tries to classify them correctly as real or fake. After training, the focus is often on the generator, which can be used to produce novel results (i.e. faces, text, and genetic data). Typically the generator is itself a neural network operating on random noise but in `pg-gan` the generator is instead an evolutionary model. The fitted generator from `pg-gan` is therefore directly interpretable in terms of population genetic parameters. Because the specification of such models is relatively standardized (population size changes, splits, migrations, admixtures, etc.), we can easily use the resulting model in downstream applications or make it available to other researchers in a standard format (see Adrion 2020). For example, we could use the demography as a null model or as training or validation data for a second machine learning method.

Although well-trained discriminators can be used downstream to validate the examples created by other generators (e.g. Salimans *et al.* 2016), there is typically less emphasis on using or analyzing the trained discriminator. In this work, we develop a new approach that uses the trained discriminators of `pg-gan` to identify non-neutral regions. The major advantage of this approach over existing approaches is that it requires very few simulated selected regions during training. Despite recent advances (e.g. Haller and Messer 2019), simulating realistic natural selection is very difficult, with many parameter choices that make mirroring real data challenging. These include the type of selection, time of onset of selection, selection strength, and frequency of the selected variant at the present. For models with exponential growth or large population sizes, forward simulations become prohibitively computationally expensive. Our approach bypasses the need to simulate large numbers of selected regions by instead detecting regions of real data that do not conform to an inferred (neutral) demographic model. This approach is similar to traditional population genetic approaches to detecting selection by looking for regions that are outliers in terms of divergence, diversity, or haplotype structure (Akey 2009). However, instead of looking at one or a combination of summary statistics (Grossman *et al.* 2013; Sugden *et al.* 2018), our approach operates directly on the observations, and in principle is able to use all the information present in the data.

Understanding what neural networks actually learn has been a major goal of interpretability work for many years. Despite progress in the image domain (e.g. Simonyan *et al.* 2013; Ribeiro *et al.* 2016; Montavon *et al.* 2018), less is known about what CNNs for population genetic data are learning (see Nait Saada *et al.* 2021; Cecil and Sugden 2023 for recent progress). Our interpretability analysis links traditional summary statistics (which are known to be informative for selection and other evolutionary parameters) with the hidden nodes of the discriminator network. This gives us a window into the network's ability to compute existing summary statistics, importance and redundancy of different statistics, and differences between discriminators.

Overall our method represents a novel approach to detect natural selection and other types of non-neutrality, while avoiding both demographic parameter mis-specification and extensive selection simulations. Our software is available open-source at https://github.com/mathiesonlab/disc-pg-gan.

## Materials and methods
### GAN training
Our existing GAN, `pg-gan`, consists of a CNN-based discriminator and an `msprime`-based generator that are trained in concert. During training, the generator selects parameters to use for the simulation software `msprime` (Kelleher *et al.* 2016; Baumdicker *et al.* 2022), creating "fake" regions. The discriminator learns to distinguish these simulated neutral regions from the real training data, which contain both neutral and selected regions. With feedback from the discriminator, the generator learns which parameters will produce the most realistic simulated data, and the discriminator gets better at picking up on differences between the two types of data. `pg-gan` is implemented in `tensorflow` (Abadi *et al.* 2015) and can be run on a graphics processing unit using `CUDA` (NVIDIA and Fitzek 2020). As the discriminator is a CNN, we are able to train it using traditional gradient descent approaches (i.e. backpropagation). However, since we currently cannot pass gradients through the coalescent with recombination, we train the generator using a gradient-free approach (in this case simulated annealing, but other methods are possible). We typically use up to 1.5 million simulated neutral regions to train `pg-gan`. At the end of this optimization process, we obtain a generator that is good at creating realistic simulated data, and a discriminator that can identify real data accurately (Wang *et al.* 2021).

Here, we focus on the trained discriminator rather than the generator. We modified `pg-gan` (Wang *et al.* 2021) to save the trained discriminator network (original architecture from Chan *et al.* 2018). The input to this CNN is a genomic region represented by a matrix of shape $(n, S, 2)$ where $n$ is the number of haplotypes, $S$ is the fixed number of consecutive SNPs, and there are two separate channels: one for the haplotype data and one for the inter-SNP distances (duplicated down the rows). The intuition behind including inter-SNP distances is that it provides a measure of SNP density, which is indicative of both demography and selection. The genotype and distance channels are analogous to the three RGB (red, green, blue) color channels for images.

For one-population evolutionary models, the CNN has two convolutional layers with filters of size $1 \times 5$, which maintains permutation-invariance across the haplotypes. We use ReLU activation functions and $1 \times 2$ max-pooling after each convolutional layer, then apply a permutation-invariant function (in this case `sum`) to distill information along the haplotype dimension. Finally we include two fully connected layers and then output a single scalar value (probability of the example being real). The convolutional layers have 32 and 64 filters (respectively) and the fully connected layers have 128 nodes each, for a total of 76,577 trainable parameters. To optimize these weights, we use binary cross-entropy as the loss function, with Adam optimization. The CNN architecture itself is not optimized during training, although it is possible this could improve results for other applications. Due to the stochastic nature of GAN training, we independently train `pg-gan` 20 times and combine the results into an ensemble model described below.

### Discriminator training failure
In a small number of cases, GAN training (which is often subject to difficulties due to the minimax nature of the optimization problem) fails. In all our experiments, this manifested as discriminators predicting the same value for all regions (i.e. the network did not learn anything). We therefore discard these discriminators from further analysis.

### Discriminator fine-tuning and prediction
The discriminator output (probability that the region is real) can be thought of as analogous to a *p*-value, in the sense that it measures the probability of an observation under a particular (demographic) model. Since the real data, unlike the simulated data, include regions that have experienced selection, the intuition is

that regions that the discriminator predicts to be real with high probability correspond to non-neutral regions, since those have characteristics found in real but not simulated data. One concern for using these predictions to detect selection is that there are neutral processes that are found in the real but not simulated data (biased gene conversion, heterogeneity in mutation rates, genotype or reference errors, and so on) which might be identified by the discriminator. We therefore incentivize the discriminator to focus on a particular mode of selection by *fine-tuning* it using a small number of forward simulations.

Specifically, we consider three types of selection: positive selection, balancing selection, and adaptive introgression. In all cases we use a modest number of regions (3,000 neutral regions and 2,400 selected regions), with 20% of the total reserved for validation. The discriminator is trained on the remaining 80% for 2,000 mini-batches, each of size 50 (roughly 24 epochs). We again use binary cross-entropy as the loss function, except now a label of 0 corresponds to neutral regions and a label of 1 corresponds to selected regions. This procedure *fine-tunes* the weights of the network such that high discriminator predictions can be more reliably interpreted as evidence for particular modes of selection (see Section "Non-neutral simulations for fine-tuning" below for more details of the simulations used for fine-tuning). The fine-tuned discriminator can then be used to predict outcomes for new genetic regions that were not observed during training (often from a closely related but distinct population). A schematic of the entire workflow can be found in Fig. 1.

We use receiver operating characteristic (ROC) curves to evaluate the improvement in performance created by fine-tuning (see Fig. 3 and Supplementary Figs. S3 and S4a). Notably, in all cases of failed training, fine-tuning also fails to improve the discriminator—prediction results are the same before and after. Additionally, we compare randomly initialized CNNs (mirroring the discriminator architecture) and trained discriminators as they are fine-tuned, which allows us to measure the impact of the initial pg-gan training. Below we describe algorithm details and validation approaches, as well as an ensemble approach for successfully trained discriminators.

## Population data and demographic models

To ensure that we are training and testing on different datasets, we use three pairs of similar populations from the 1,000 Genomes Project (1000 Genomes Project Consortium 2015): CEU and GBR (Northern European), CHB and CHS (East Asian), and YRI and ESN (West African). These pairs were selected since they are the most similar in terms of genetic distance (see 1000 Genomes Project Consortium 2015, Fig. 2). Specifically, we train discriminators using CEU, CHB, and YRI, then test on GBR, CHS, and ESN, respectively. Although the number of samples may differ between the populations, the permutation-invariant architecture of the discriminator alleviates any dimension mismatches. This is because after the convolutional layers, the permutation-invariant function collapses information across the haplotypes into a single value (i.e. with the sum function). If the number of haplotypes differs greatly between train and test populations, this will cause discrepancies in the final network output, but 198 vs 182 haplotypes is negligible. Population sample sizes and number of regions are given in Table 1.

For all populations, we used an exponential growth model and fitted five parameters using pg-gan: the ancestral population size ($N_1$), the timing of a bottleneck ($T_1$), the bottleneck population size ($N_2$), the onset of exponential growth ($T_2$), and the exponential growth rate ($g$). For all simulations, we used a constant mutation rate of $\mu = 1.25 \times 10^{-8}$ per base per generation and recombination rates drawn from the HapMap recombination map (International HapMap Consortium 2007). We simulated $L = 50$ kb regions and then extracted the middle $S = 36$ SNPs. To avoid assuming an ancestral state, for each SNP we encode the minor allele as 1 and the major allele as −1. We found that centering the data around zero worked best with typical neural network activation functions, and also allowed us to zero-pad a small number of regions with fewer than $S$ SNPs. For real data, we filter non-biallelic SNPs and ensure that at least 50% of the bases are inside callable regions (see 1000 Genomes Project Consortium 2015), so that the discriminator avoids making real vs fake distinctions based on artifacts.

## Non-neutral simulations for fine-tuning

To create the non-neutral simulations used for fine-tuning, we use SLiM (Haller and Messer 2019; Haller *et al.* 2019), which can model a variety of types of natural selection. For the demography, we use the same evolutionary model (exponential growth) as we used during pg-gan training. For each population, we use the parameters shown in Table 2, which were obtained from a run

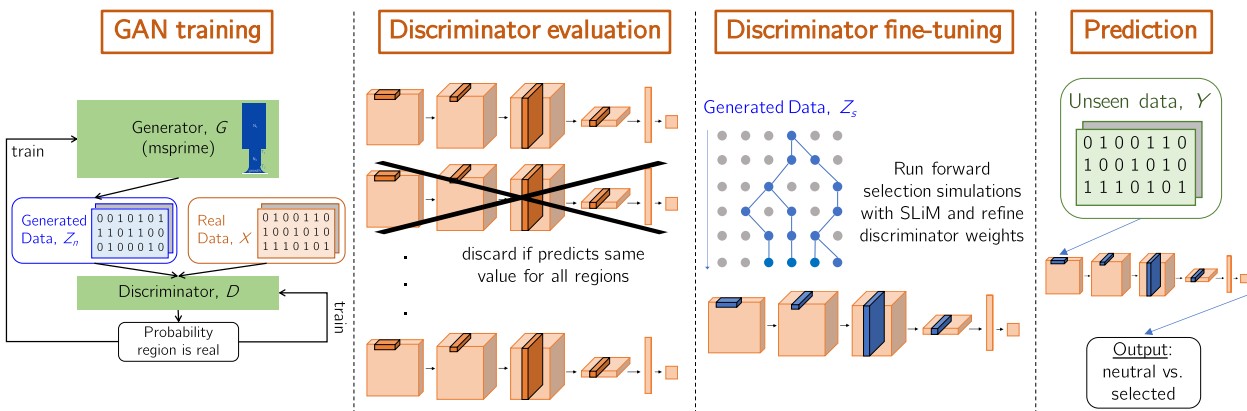

**Fig. 1.** Selection inference workflow. First we run pg-gan 20 times to obtain a collection of discriminators. In the evaluation step, we discard any discriminator that does not pass quality checks (i.e. predicting the same value for every region). Retained discriminators (roughly 85%) are fine-tuned with a modest number of non-neutral simulations. Finally, real data from a held-out population with similar ancestry are fed through each fine-tuned discriminator to obtain a probability. Regions where a large fraction of the discriminators produced a high probability are candidate selected regions.

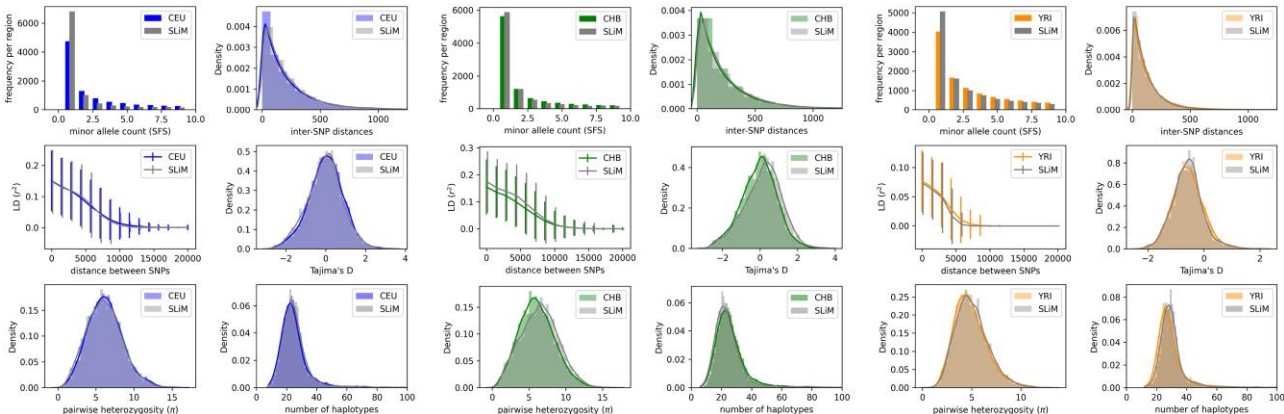

**Fig. 2.** Summary statistic comparison of real training data (CEU, CHB, and YRI) to data simulated under SLiM with parameters inferred by pg-gan. The number of regions used for comparison was 5,000.

of pg-gan with CEU, CHB, or YRI training data, respectively. During this training run, we capped the exponential growth parameter at 0.01, as allowing too much growth led to very slow SLiM simulations. Additionally, we capped $T_2$ (the onset time of exponential growth) at 750 generations for YRI. A visualization of the match between real data and simulated data summary statistics is shown in Fig. 2.

Under these demographic models, we produced three different types of fine-tuning datasets:

- Positive selection from a new mutation
- Balancing selection (over-dominance)
- Adaptive introgression (using the demographic model from Gower *et al.* 2021)

For simulations of positive selection, selection is introduced by adding the beneficial mutation to one individual in the population, 1,000 generations before the present. The mutation is added to the center of the region. If the mutation is lost in any generation, we restart the simulation from the previous generation until we have reached the present—the resulting tree sequence can then be stored for further processing. First, the tree sequence is recapitated using coalescent simulations for efficiency (using the pyslim module). This step ensures that we have a complete tree sequence with all necessary ancestral nodes. Next, we sample the number of haplotypes to match the training population sample size. This allows us to prune and simplify the tree sequence before adding neutral mutations with msprime. For each region we store arrays with S SNPs, as well as arrays with all the SNPs, in order to compute Tajima's D. We simulate 600 regions for each of the selection coefficients $s = 0.01, 0.025, 0.05, 0.1$, as well as 3,000 neutral regions.

For simulations of balancing selection, we use the same procedure as the positive selection simulations, except using a dominance coefficient of $h = 2.0$ (i.e. heterozygote advantage), loosely following code from Isildak *et al.* (2021). We again simulate non-neutral regions using a variety of selection coefficients, to contrast with the same neutral regions as above.

For simulations of adaptive introgression, we use the human/Neanderthal split model from Gower *et al.* (2021) which includes African, European, and Neanderthal haplotypes. We retain only the European haplotypes (representing CEU) and process them in the same fashion as above (i.e. S biallelic SNPs as the first channel and inter-SNP distances as the second channel). For both balancing selection and adaptive introgression, we restrict our

analysis to the CEU training population and GBR testing population.

## Validation on simulated and real data

To validate predictions of positive selection after fine-tuning, we examine each discriminator's performance on both simulated and real data. For validation with simulated data, we held aside SLiM simulations under positive selection and ran them through each discriminator. For real data, we used genomic regions previously identified as targets of selection by Grossman *et al.* (2013) using a combination of different summary statistics. We use the 153 regions from their Supplementary Table S1, including genes involved in metabolism, disease resistance, and skin pigmentation. We converted the start and end positions of each region to hg19 coordinates and sorted by population (CEU, CHB/JPT, and YRI). This resulted in a comparison between four types of data:

- Regions simulated using msprime under the demographic parameters inferred by pg-gan (corresponding to the current discriminator)
- Regions simulated using SLiM under various selection strengths
- Regions under positive selection as identified in Grossman *et al.* (2013)
- All other real regions (mostly neutral)

For balancing selection and adaptive introgression, we found very little overlap between published regions of balancing selection identified using different methods (i.e. Siewert and Voight 2017; Bitarello *et al.* 2018), or for Neanderthal adaptive introgression results (i.e. Sankararaman *et al.* 2016; Setter *et al.* 2020). Thus for these two types of non-neutrality, we validated using simulations only.

## Discriminator ensemble method to detect selected regions

To assess the discriminator's predictions genome-wide, we iterate through the entire genome (using non-overlapping 36-SNP windows) and make a prediction for each region, then smooth the results by averaging probabilities in five consecutive windows. Although we refer to the predictions as probabilities of selection, we caution that these values should not be over-interpreted as well-calibrated or posterior probabilities. To visualize the results, we create modified Manhattan plots by plotting the probabilities on the y-axis (on a log scale). Individual discriminators can vary

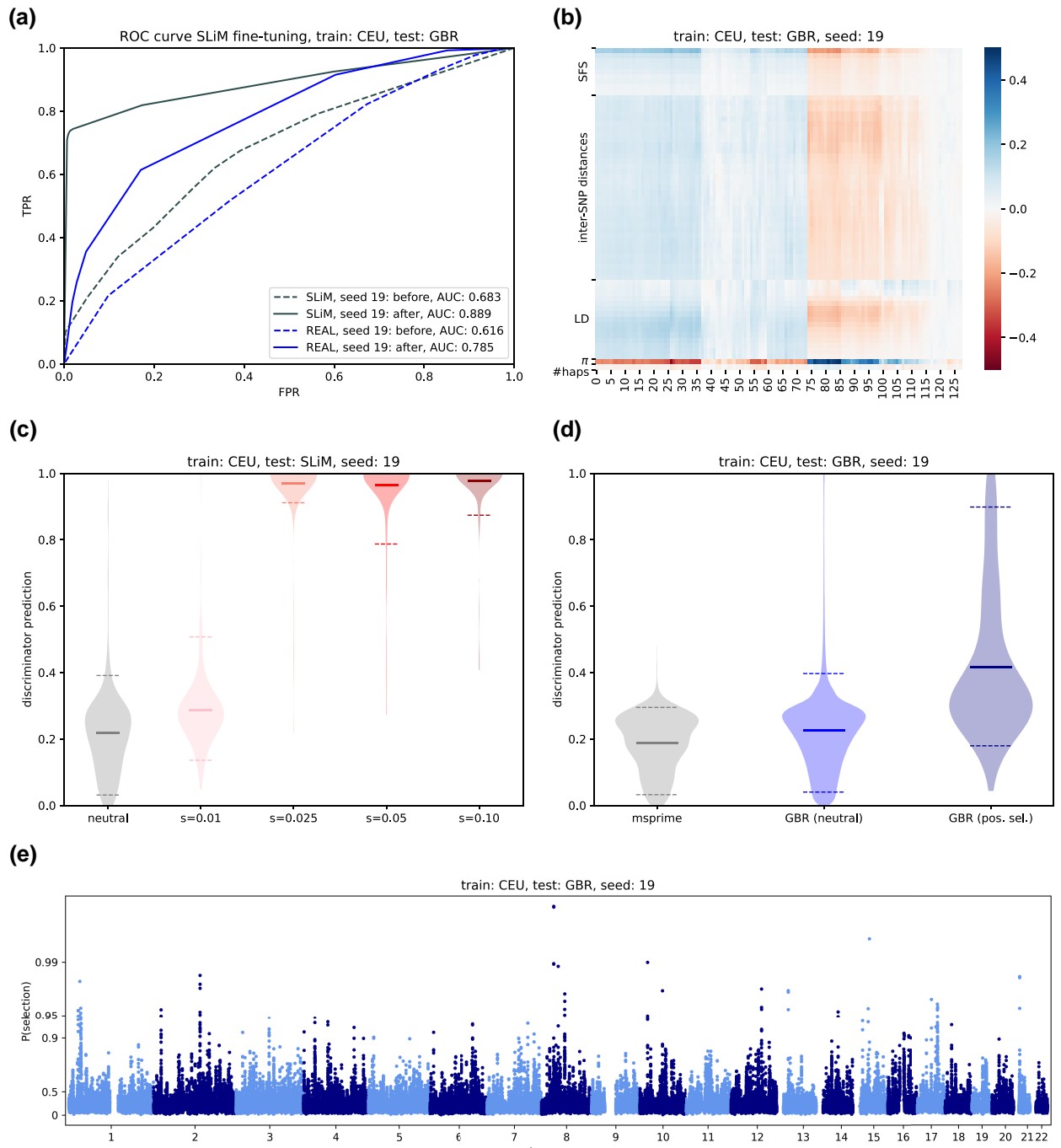

**Fig. 3.** CEU-trained discriminator (seed 19), fine-tuning with positive selection simulations, and test data from GBR. a) ROC curve showing performance on selected regions before and after fine-tuning. All predictions (both simulated and real) are made for unseen (test) data. SLiM indicates simulated data with various selection coefficients and REAL indicates selected regions from Grossman *et al.* (2013). b) Correlation heatmap between discriminator hidden units (x-axis) and classical population genetics summary statistics (y-axis). The columns (hidden units) were clustered according to their similarity in terms of summary statistic correlation profiles. c) Performance of discriminator on unseen simulated data under various selection strengths. d) Performance of discriminator on selected regions from Grossman *et al.* (2013). e) Genome-wide Manhattan plots of discriminator predictions on real test data.

in their performance, so we also created an ensemble classifier by averaging predictions for each region over all successfully trained discriminators.

## Interpretability analysis

Finally, we sought to understand what the discriminator network is learning and connect its output to our biological intuition. We investigated whether the network had learned how to compute summary statistics that are known to be informative for evolutionary parameters (demography, selection, etc.). To this end, we computed the correlation between various summary statistics and parts of our discriminator network: the node values of the last hidden layer and the final prediction. The motivation for analyzing the last hidden layer is that deep into the network, information

**Table 1.** Populations analyzed along with the number of individuals in each population.

| Code | Population description | # Individuals | # Regions |
|------|------------------------|---------------|-----------|
| CEU | Utah residents with Northern and Western European ancestry | 99 | 292,936 |
| GBR | British in England and Scotland | 91 | 277,038 |
| CHB | Han Chinese in Beijing, China | 103 | 280,215 |
| CHS | Han Chinese South | 105 | 270,351 |
| YRI | Yoruba in Ibadan, Nigeria | 108 | 473,493 |
| ESN | Esan in Nigeria | 99 | 460,180 |

*Note.* The number of regions is determined using non-overlapping 36-SNP windows, where at least 50% of bases must be in the accessibility mask.

**Table 2.** Parameters of the exponential growth model, inferred for each population through a training run of pg-gan.

| Population | $N_1$ | $N_2$ | $g$ | $T_1$ | $T_2$ |
|------------|-------|-------|-----|-------|-------|
| CEU | 22,552 | 3,313 | 0.00535 | 3,589 | 1,050 |
| CHB | 24,609 | 3,481 | 0.00404 | 4,417 | 1,024 |
| YRI | 23,231 | 29,962 | 0.00531 | 4,870 | 581 |

*Note.* These point estimates are used for the demography in our SLiM selection simulations.

from the haplotypes should be greatly distilled and processed, with informative high-level "features" extracted. The remaining step in the network is a linear combination of these hidden node values to create a logit value, which can be converted into a probability. This probability may also be correlated with summary statistics, as after fine-tuning it represents the network's prediction of neutral vs selected. Indeed, this value is itself a very complicated summary statistic. We computed pairwise correlations between 128 hidden values (plus the final probability) and the following 61 summary statistics:

- The first nine entries of the folded site frequency spectrum (SFS), excluding non-segregating sites.
- The 35 inter-SNP distances between the 36 SNPs of each region. These distances are rescaled to fall between 0 and 1. (Note that this vector is also an input to the network via the second channel.)
- Fifteen linkage disequilibrium (LD) statistics (see Wang *et al.* 2021 for details of the computation).
- Pairwise heterozygosity ($\pi$).
- The number of unique haplotypes across the 36 SNPs.

For the correlation with the final probability, we visualized the results as a bar chart for each summary statistic. For the last hidden layer, we calculated Pearson correlation coefficients and plotted them in a heatmap with summary statistics along rows and hidden values along the columns. To better visualize relationships between the hidden units, we clustered the columns according to their similarity using agglomerative clustering. The final visualization groups hidden nodes that are performing similar computations—since we perform a dropout (with rate 0.5) after each fully connected layer, we expect some redundancy in node behavior. We experimented with reducing the number of hidden units in the last few layers, as well as adding three fully connected layers (instead of two). However, all these modifications to the discriminator CNN resulted in degenerate training results (e.g. predicting all regions as real) so we pursued only the original architecture.

**Algorithm 1.** Ensemble method for using pg-gan discriminators to detect non-neutral regions

**High-level idea:** run pg-gan several times on the train population to obtain a set of discriminators. Retain those trained successfully and use them in an ensemble to identify candidate selected regions in the test population.

$\Theta$—parameters for an evolutionary model
$X$—real training data ( $X_n$ neutral, $X_s$ known selected)
$Y$—real test data ( $Y_n$ neutral, $Y_s$ known selected)
$Z$—simulated data ( $Z_n$ neutral, $Z_s$ under selection)

**Input:** $K$ discriminators ( $D_1, \ldots, D_K$) trained with $X$
**Output:** successful discriminators $C$, predictions for all regions in $Y$

**for** each discriminator $D_k$, generator $G_k$, inferred parameters $\Theta_k$ **do**
  Let $Z_n = G_k(\Theta_k)$ be neutral data simulated under $\Theta_k$

  **if** $D_k$ does not predict the same value for all regions **then**
    Fine-tune $D_k$ using selection simulations $Z_n$ (label 0) and $Z_s$ (label 1)
    Add $D_k$ to set of successfully trained discriminators $C$
  **end**
**end**
**for** region $y$ in $Y$ **do**
  **for** discriminator $D_i$ in $C$ **do**
    $p = D_i(y)$
  **end**
  Average all predictions—if the resulting probability exceeds a threshold (say 0.75), return as a putatively selected/non-neutral region
**end**

## Results
### Summary of discriminator training

For each training population (CEU, CHB, and YRI), we ran pg-gan 20 times to obtain a set of 60 discriminators. For each run we used a different seed, although even runs with the same seed can produce different results depending on the version of tensorflow and CUDA. In 10 cases, training failed and the resulting discriminator predicted the same value for all regions (in other words, it ignores the input data—see Supplementary Figs. S1 and S2 for an example). After we discarded these discriminators, we were left with 50 (17 for CEU, 18 for CHB, and 15 for YRI). Below we discuss the outcomes of fine-tuning and prediction with three representative discriminators (one for each population pair), then explain how we used all successfully trained discriminators in an ensemble method to predict positive selection.

### Fine-tuning and validation on simulated data

In Fig. 3a and Supplementary Figs. S3 and S4, we show the impact of fine-tuning using positive selection simulations on individual discriminators (seed 19 for CEU, seed 6 for CHB, and seed 10 for YRI). In the ROC curves for simulated data, a *positive* represents a region simulated under any strength of selection, and a *negative* represents a neutral region. For *real* neutral vs selected data, we use the regions identified in Grossman *et al.* (2013) as the truth set. Dashed lines represent discriminator predictions before fine-tuning and solid lines represent predictions after fine-tuning. Fine-tuning improved discriminator predictions in all cases, although in a few cases the improvement was marginal. For YRI-trained discriminators tested on ESN, improvements were generally more marginal than for CEU- and CHB-trained discriminators.

After fine-tuning, we run selection simulations (again created with SLiM) through each discriminator. In Fig. 3b and

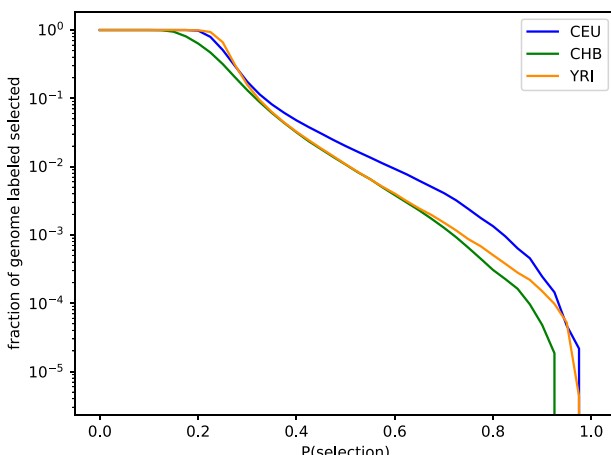

**Fig. 4.** Ensemble probability results for positive selection. On the x-axis is the probability of selection, and the y-axis shows the fraction of the genome classified as selected if we used the given probability threshold (log scale). Across populations the average probability of selection is 0.26, and approximately 1.4% of the genome has a probability of selection >50%.

Supplementary Figs. S3 and S4, we visualize these results through violin plots with 1,000 neutral regions and 100 regions of each selection strength (all unseen during fine-tuning). Most discriminators group neutral regions and those with selection strength $s = 0.01$ together, with probabilities around 0.2–0.3. Regions under stronger selection have much higher probabilities (around 0.9–1). Although the network was fine-tuned with only a few simulated regions under selection, the real data include regions of selection, so this general pattern exposes the ability of most discriminators to see selection as realistic.

### Validation on known selected regions

To understand how the discriminator behaves when presented with real data from a test population, we produced predictions for regions known to be under selection (from Grossman *et al.* 2013) and compared to predictions from the rest of the genome. We also included data simulated under the demography inferred by the pg-gan training run that produced the given discriminator. In theory, these are the data which the discriminator thinks are most similar to real data.

Overall, we find large differences in the predicted probabilities for selected vs neutral regions (Fig. 3 and Supplementary Figs. S3 and S4c). For example, in Fig. 3, we show the performance of a CEU-trained discriminator (seed 19) on real data from GBR. Horizontal bars represent means (solid) and 0.05–0.95 quantiles (dashed). In this case, the *t*-test *p*-value for a difference in means between selected (0.423) vs neutral (0.261) probabilities is $1.63 \times 10^{-50}$. Of the 50 successfully trained discriminators, the *p*-value is significant (at the 0.05 confidence level) in all cases. Predictions for simulated data (neutral) and neutral real data typically have very similar distributions, indicating that the generator's demographic model trained successfully. In simulated data, predictions are better for YRI demography, compared to CEU or CHB, but for real data this pattern is reversed. We suspect that power is theoretically higher for YRI due to higher diversity, but that the selection signals from Grossman *et al.* (2013) are either less reliable or less shared between YRI/ESN (compared to CEU/GBR or CHB/CHS).

For each discriminator, we also performed a genome-wide scan over real test regions (see Fig. 3 and Supplementary Figs. S3 and

S4e). In the case of CEU (seed 19) for example, we computed predictions for GBR data in 36-SNP regions, then averaged 5 such regions (in overlapping windows) to create the final predictions for these modified Manhattan plots. Similarly, we ran CHS through the CHB-trained discriminators (seed 9 shown) and ESN through the YRI-trained discriminators (seed 4 shown).

### Ensemble results

For the successful discriminators of each training population, we average the probabilities for each test region to create ensemble results. To smooth the results, we further average predictions over five consecutive 36-SNP regions. In other words, the smoothed prediction for a single 36-SNP region is the average of itself, along with two downstream and two upstream regions. Across populations, the mean probability of selection is 0.26, and approximately 1.4% of the genome has a probability of selection >50% (2% for GBR, 1% for CHS, and 1% for ESN; Fig. 4). We confidently identify well-known targets of selection, including *LCT* (probability of selection, $P = 0.86$; Bersaglieri *et al.* 2004), *SLC24A5* ( $P = 0.96$; Lamason *et al.* 2005), *KITLG* ( $P = 0.87$; Field *et al.* 2016; Stern *et al.* 2019), and *OCA2/HERC2* ( $P = 0.92$; Lao *et al.* 2007) in GBR; *EDAR* ( $P = 0.56$) in CHS (Voight *et al.* 2006); and *APOL1* ( $p = 0.79$; Genovese *et al.* 2010) in ESN. We also assign high probabilities of selection to several potentially novel targets, including *APPBP2/PPM1D* ( $P = 0.95$) in GBR; *CENPW* ( $P = 0.91$) and *EXOC6B* ( $P = 0.92$) in CHS; and *EHBP1* ( $P = 0.96$), *WHSC1L1* ( $P = 0.96$), and *ARID1A* ( $P = 0.99$) in ESN (Fig. 5). Full predictions are given in Supplementary Table S1.

To more systematically validate our results, we obtained selected regions from Grossman *et al.* (2013) (Supplementary Fig. S6), which uses a composite of multiple classical selection statistics to identify selected regions. We calculated our mean prediction probabilities inside and outside these regions. Then we performed permutation testing by shuffling the locations of the selected regions from Grossman *et al.* (2013) randomly throughout the genome, while still retaining the original length of each region. For each of 100 such permutations, we re-computed the mean prediction probabilities inside and outside the permuted regions. In all cases, the mean selection probability of the true known selected genes ( $D(Y_s)$) was much higher than the permuted values. *P*-values were $\leq 10^{-324}$ for all populations and our mean was 8–12 standard deviations above the permutation testing mean (Table 3).

### Comparison with randomly initialized networks

To assess the impact of pg-gan training plus fine-tuning vs fine-tuning only, we compared trained discriminators with randomly initialized CNNs with the same architecture (in other words, these networks look like discriminators before pg-gan training begins). As both types of networks are fine-tuned with selection simulations, we observe that the loss functions eventually reach roughly the same value (see Fig. 6a), but the trained discriminator loss function starts out much lower. For simulated data, the randomly initialized network still obtains a high accuracy through fine-tuning (AUC, area under the curve = 0.867 in Fig. 6b vs AUC = 0.889 in Fig. 3a). However, for the real validation data, the pg-gan trained discriminators are much more successful at identifying real signals of selection (AUC = 0.626 in Fig. 6b vs AUC = 0.785 in Fig. 3a). This suggests to us that one of the main benefits of the GAN training is to improve the robustness of the fine-tuned discriminator to systematic differences between real and simulated data (for example, the consequences of biological processes that are not included in the simulations).

**(a)**

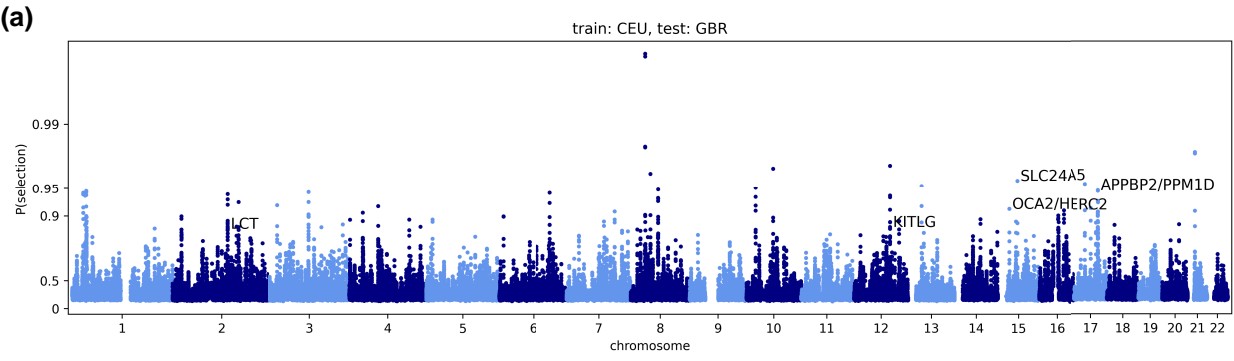

**(b)**

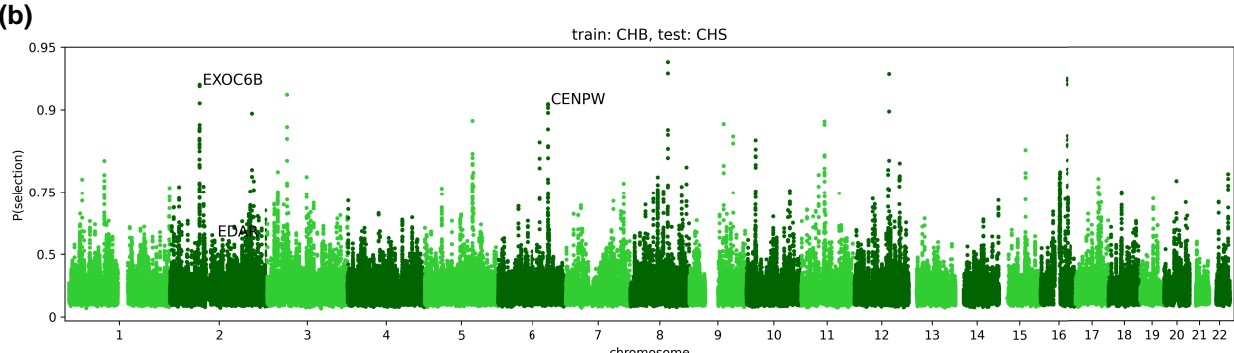

**(c)**

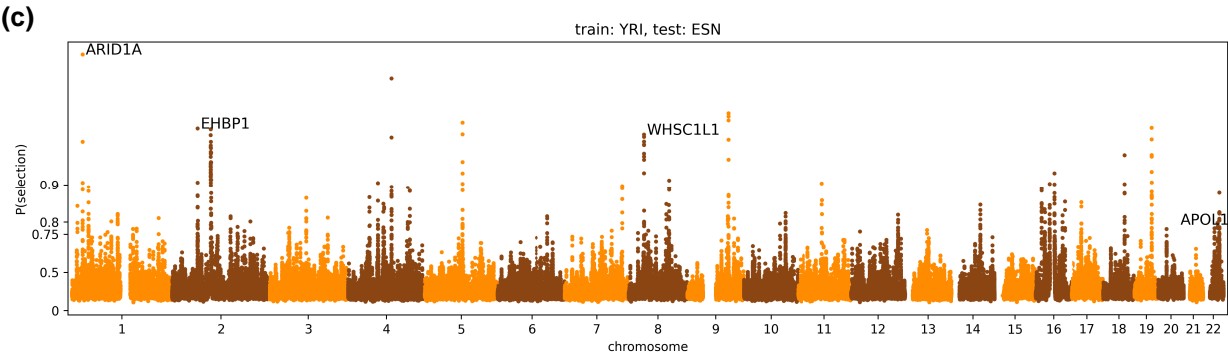

**Fig. 5.** Ensemble results for positive selection. Genome-wide selections scans for a) CEU/GBR, b) CHB/CHS, and c) YRI/ESN. In each case, the *x*-axis represents genomic position, the *y*-axis represents the probability of selection (log scaled), and each point represents the average of five consecutive 36-SNP windows. In black text we highlight known and novel regions with a high probability of selection.

**Table 3.** Ensemble results for positive selection.

|  | CEU/GBR | CHB/CHS | YRI/ESN |
|---|---|---|---|
| # Successful discriminators | 17/20 | 18/20 | 15/20 |
| Mean neutral $D(Y_n)$ | 0.269 | 0.233 | 0.272 |
| Mean selection $D(Y_s)$ | 0.421 | 0.377 | 0.396 |
| *P*-value for difference in means | $8.52 \times 10^{-43}$ | $2.31 \times 10^{-94}$ | $9.52 \times 10^{-74}$ |
| # Std dev above mean (permutation testing) | 8.41 | 9.75 | 11.71 |

*Note.* For each population, we report the mean predicted probabilities for regions that are selected vs neutral (according to Grossman *et al.* 2013), and the *P*-value for the difference in mean probabilities between these two sets of regions.

## Fine-tuning with different types of selection

For the training population CEU, we fine-tuned the same 17 discriminators with 2 additional types of selection: balancing selection and adaptive introgression. The results for seed 19 are shown in Fig. 7 for simulated testing data. For balancing selection, we find

that fine-tuned discriminators are easily able to detect this signal (AUC = 0.921), while adaptive introgression is a more difficult task (AUC = 0.62). We hypothesize that adaptive introgression methods really benefit from the inclusion of haplotypes from the source population and an outgroup population. Using CEU haplotypes only (as well as a modest number of SNPs per region) likely diluted this signal.

## Interpretability analysis

After performing correlations between the hidden units of each discriminator and traditional summary statistics (both calculated on the test data), we find some common patterns. Our hierarchical clustering approach shows extensive redundancy in the hidden units, which is to be expected given the dropout computations included in the last fully connected layers. We note that none of the correlation values are particularly high—the largest (in magnitude) are around ±0.4. Frequently the highest correlations are with rare variants (first entries of the SFS), LD statistics, and $\pi$

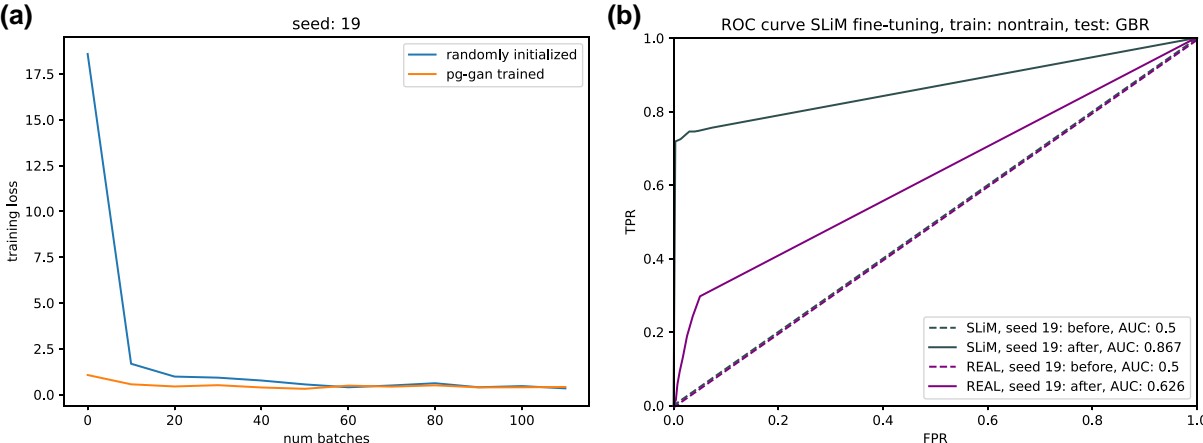

**Fig. 6.** Randomly initialized discriminator network comparison for seed 19 (positive selection case). a) Loss curves for randomly initialized vs `pg-gan` trained discriminator networks, as they are fine-tuned with batches of selection simulations. b) Performance of randomly initialized network after fine-tuning, on both simulated and real regions. For simulated data the classification accuracy is still high, but substantially worse for real data as compared with Fig. 3a.

(pairwise heterozygosity). For some discriminators, inter-SNP distances also seem to be important. In Fig. 3 and Supplementary Figs. S3 and S4d, we show example heatmaps of correlation values between hidden values of the last layer and summary statistics. In Supplementary Fig. S5, we show correlation results with the final probability, for all selection strengths in the positive selection case. We find that for selection coefficients greater than 0.01, a pattern emerges with strong correlations between the networks output and the number of singletons and $\pi$. LD statistics are also correlated, with LD for closer pairs of SNPs negatively correlated and LD for further pairs of SNPs positively correlated. Overall, the correlation values were higher for the final probability than for the last hidden layer. We also investigated correlation patterns between summary statistics and the final probability in the balancing selection case, since the validation results were promising. However, we did not see a clear correlation pattern, perhaps indicating that these standard summary statistics are not informative for detecting balancing selection.

## Runtime analysis

Overall, the runtime of our method is dominated by the initial training of `pg-gan`, which takes 5–7 h per discriminator. After the initial training, fine-tuning is very fast, about 1 min per discriminator. Selection simulations with `SLiM` were completed in advance of fine-tuning and were highly sensitive to the demography used to mirror each population. YRI simulations took about 4–5 h per 100 regions, CHB simulations took about 1 h per 100 regions, and CEU simulations took about 4–8 h per 100 regions, depending on the strength of selection (all simulations were parallelized). We hypothesize that CEU took the longest due to the highest inferred exponential growth rate. CHB had the lowest average effective population size, likely resulting in faster simulations.

In terms of the computational saving of pre-training, the loss functions achieve roughly the same value after 20 batches. With 50 regions per batch, this represented 1,000 SLiM simulations. Although we perform SLiM simulations in advance and they can be parallelized, this still represents potentially 40–80 h for CEU. Unless a substantial number of cores are used, this implies there are usually computational savings to pre-training. However, based on Figs. 3a and 6b, the more important benefit is increased robustness and accuracy for real data.

## Discussion

This work develops three novel aspects of machine learning in population genetics. First, we show how a discriminator trained as part of a GAN can be used as a classifier independent of the generator. Second, we show how the discriminator can be further incentivized to give high probabilities to selected regions in particular through fine-tuning on a small number of selection simulations. Finally, we show how the hidden units of the discriminator can be interpreted in terms of known summary statistics. In particular, we find that rare variant counts, LD statistics, and pairwise heterozygosity are correlated with hidden units in the discriminator network.

A major advantage of our approach is that we do not need to simulate large numbers of selected regions across a high-dimensional space of selection parameters. Training `pg-gan` is not instantaneous, but is much faster than it would be if we used selection simulations throughout the entire training procedure. A typical training run of `pg-gan` uses around 1.5 million neutral simulated regions depending on the number of demographic parameters. In contrast, we only use around 5,000 selection simulations for fine-tuning. By adding this step, we are able to quickly create discriminators that have the ability to pick up on real selected regions. Additionally, although randomly initialized discriminators are able to identify *simulated* selected regions fairly well after fine-tuning, their results are much worse (than pre-trained networks) for *real* data. This suggests that GAN pre-training with real data makes the resulting discriminators more robust to differences between real and simulated data that are potentially unrelated to natural selection.

One advantage of our approach is that the evolutionary generator is highly customizable in a biologically meaningful way. If it were completely unconstrained (say a CNN with many parameters), then the generator could learn to mimic all types of real data, including neutral regions and those under various forms of selection. By constraining the generator to produce neutral data only, it learns a null model that best approximates the majority of real data regions. Regions that do not conform are therefore more easily identified as under selection. One caveat is that if selection is highly prevalent in a species or population, GAN training may incorrectly infer a bottleneck or other demographic scenario often

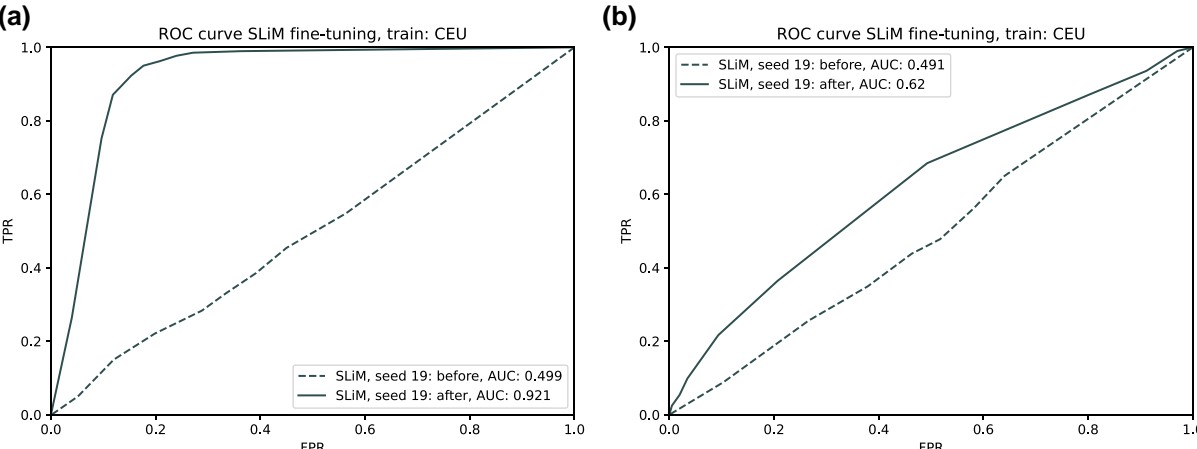

**Fig. 7.** Balancing selection and adaptive introgression results. a) Performance of `pg-gan` trained discriminator (seed 19) after fine-tuning with balancing selection simulations. b) Performance of the same discriminator after fine-tuning with adaptive introgression simulations.

confused with selection. Our approach is conservative in that it tries to explain as much of the variation as possible using neutral processes and may be inaccurate if a large fraction of the variation is attributable to selection.

In order to validate our approach in other species, it would be helpful to identify a set of regions under positive selection (for testing). This could be bypassed with high-quality selection simulations, but these might be difficult to obtain depending on the species. Here, we used known selected regions to validate our approach for positive selection but an alternative would be to instead incorporate these into `pg-gan` training as an additional fine-tuning step. Lack of reliable ground truth data for other forms of selection limits our ability to validate these approaches. Currently, our interpretability work is on a global scale, meaning that we analyze genome-wide patterns. Future work could examine why the discriminator makes predictions for particular regions, especially outliers with high probabilities.

A caveat of our approach is that the non-neutral regions identified by the discriminator cannot be directly interpreted in terms of selection parameters. Simulations suggest that we can easily detect hard sweeps with selection coefficients greater than around 1%, but may not be able to detect weaker selection (similar to state-of-the art population genetic methods; Field *et al.* 2016; Palamara *et al.* 2018). One potential way to obtain this information could be to still use the pre-trained discriminator, but during fine-tuning change the output to the selection coefficient, thus creating a regression problem. Additionally, it is possible that, despite the fine-tuning step, the regions we identify actually reflect different types of deviation from the generator model, such as heterogeneity in mutation or recombination rates, or structural variation. We did find regions where many discriminators output a high probability, but we did not see any obvious genomic features of interest (see Supplementary Table S1). To investigate these regions further, summary statistics might help untangle if these regions have some other notable feature such as high LD, excess of singletons, or an unusual SNP density. However, we note that almost all approaches to detecting to selection are sensitive to false discovery effects to some extent and our results, like those of all selection scans, should be treated as candidate regions requiring validation. A final caveat is that while we do interpret the output of the discriminator as a probability, it is not necessarily well-calibrated in any sense, nor does it represent a posterior probability. We do observe that among the real test regions, the average probability decreases after fine-tuning, perhaps correctly

indicating that a relatively small fraction of the (human) genome is under selection.

We showed how our approach can be extended to other regimes, including balancing selection and adaptive introgression. There are many other possibilities for fine-tuning with custom selection simulations—for example, background selection (or sweeps in the presence of background selection), underdominance, or temporally/geographically variable selection coefficients. We could even modify the last layer of the discriminator network to output a multi-class prediction that incorporates these different modes of selection, or include additional populations (i.e. Neanderthals, outgroups) in the input. Recombination rate variation and complex mutational signatures could also be incorporated (either into the main `pg-gan` training or into fine-tuning, depending on the simulation speed). More generally, we view our work as the beginning of an exploration of how the trained discriminator can be used in transfer learning approaches—for these and other evolutionary applications.

## Data availability

All human genomic data used in this study are publicly available (1000 Genomes Project Consortium 2015). Our software is also publicly available open-source at https://github.com/mathiesonlab/disc-pg-gan.

Supplemental material available at GENETICS online.

## Funding

SM is funded in part by a National Institutes of Health (NIH) grant R15HG011528. The content is solely the responsibility of the authors and does not necessarily represent the official views of the National Institutes of Health.

## Conflicts of interest

The author(s) declare no conflicts of interest.

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

*Editor: P. Ralph*