## [Peer Review File · Genetics]

Interpreting Generative Adversarial Networks to Infer Natural Selection from Genetic Data

Rebecca Riley, Iain Mathieson, and Sara Mathieson

NOTE: The reviews and decision letters are unedited and appear as submitted by the reviewers.

In extremely rare instances and as determined by a Senior Editor or the EIC, portions of a review may be redacted. If a review is signed, the reviewer has agreed to no longer remain anonymous.

The review history appears in chronological order.

Review Timeline:

Submission Date:	2023-07-09
Editorial Decision:	2023-08-08
Resubmission Received:	2023-11-10
Editorial Decision:	2023-12-12
Revision Received:	2024-01-15
Accepted:	2024-01-19

August 7, 2023

GENETICS-2023-306339

Interpreting Generative Adversarial Networks to Infer Natural Selection from Genetic Data

Dear Dr. Mathieson:

Two experts in the field have reviewed your manuscript, and I have read it as well. I found the approach and writeup refreshingly creative, clear and forward-thinking. As you will see, both reviewers liked it a good deal as well. While your manuscript is not currently acceptable for publication in GENETICS, we would welcome a substantially revised manuscript. Both reviewers have comments and concerns to be addressed in a revised manuscript. You can read their reviews at the end of this email.

Although the reviewers were very positive about the manuscript, Reviewer 1 (I think correctly) points out that the paper as written is halfway between an exploration of the generic "use a discriminator" approach and a description of a new method for finding hard sweeps. They provide some concrete suggestions for how to add more detail to either of those. (For instance, if this is a "new hard sweep finder" method, then it should probably be compared to an existing one.) I would hope to find some additional depth added to either one of these aspects of the paper in a revised version. Of course, other resolutions (besides those suggested by the reviewer) to this generic concern about a lack of depth are welcome. There is a fair amount of good work in here already, so I would hope that additional experiments are not too onerous. We look forward to receiving your revised manuscript. Please let the editorial office know approximately how long you expect to need for revisions.

Upon resubmission, please include:

1. A clean version of your manuscript;
2. A marked version of your manuscript in which you highlight significant revisions carried out in response to the major points raised by the editor/reviewers (track changes is acceptable if preferred);
3. A detailed response to the editor's/reviewers' feedback and to the concerns listed above. Please reference line numbers in this response to aid the editor and reviewers.

Your paper will likely be sent back out for review.

Additionally, please ensure that your resubmission is formatted for GENETICS
<https://academic.oup.com/genetics/pages/general-instructions>

Follow this link to submit the revised manuscript: Link Not Available

Sincerely,

Peter Ralph
Associate Editor
GENETICS

Approved by:
Nicholas Barton
Senior Editor
GENETICS

AE comments:

- The parallels to hypothesis testing are amusing, and are probably worth pointing out? (For instance, at least the non-fine-tuned network gives something analogous to a p-value, used as a metric of goodness-of-fit to a particular model.)
 - in the caption for Figure 3, "simulated data with $s=xx$ "... missing something?
-

Reviewer #1 (Comments for the Authors (Required)):

This study utilizes adversarial machine learning architectures (GANs), where a parameterized generator (population genetic

simulator) is pitted against a discriminator (a neural net classifier) tasked with distinguishing observed data from simulations. Prior work with GANs in population genetics (including by the authors of this paper) has largely focused on the trained generator -- e.g. for the purpose of inferring parameters in some particular evolutionary model or set of models. Here, the authors make the simple and interesting observation that the trained discriminator is useful in its own right, because it has been trained to distinguish features in actual data which are at odds with a "null model" (whatever evolutionary model is employed by the generator) without reference to a particular "alternative model".

Thus, the GAN training regime could be used to pre-train a neural net for tasks that are prohibitively expensive to simulate. So: fit a GAN where the generator is computationally efficient but does not simulate the process of interest, hook the GAN-trained discriminator up to a different loss function/classification task, and fine-tune it with a relatively small number of realistic simulations that *do* include the process of interest. In this case, "prohibitively expensive" means individual-based forward-in-time simulation, whereas "computationally-efficient" means coalescent simulation. The authors suggest that this strategy may make it possible to train neural nets for tasks / demographic settings that would otherwise be computationally infeasible. They apply this strategy to detect hard selective sweeps in human samples, and compare the results to a "truth" set that contains genes that are inferred to be targets of selection a priori (by a previous study). Finally, they correlate values in the last layer of the discriminator (e.g. "features" extracted by the neural net) to summary statistics to provide some insight into what the neural network is learning.

I agree with the core observation regarding the utility of the discriminator (gave me an 'a-ha!' moment); I think the paper's major point (GANs provide a means to bypass lots of computationally intensive training simulation) is very plausible; and the paper is clearly written despite the complexity of the inference method (there's not too much ML jargon and it should be accessible to many readers). However, in my opinion the most interesting point of the paper (a general-purpose pre-training strategy that can use fast but less realistic simulation) gets overshadowed by the sweep detection proof-of-concept (less interesting, because there's lots and lots of methods to detect sweeps, including ML ones). In particular:

1. The computational savings vs accuracy are never clearly demonstrated. The closest we get are Figs 3A,4A,5A, which compare neural net performance after pre-training and after fine-tuning. However, I don't think this gets the point across. What I'd like to see are training histories for randomly initialized networks as well as for pre-trained discriminators, when fed a series of fine-tuning simulations. Show validation loss against epoch (or alternatively, number of training simulations); I'd think it'd be very clear that the pre-trained CNNs have a huge advantage over the "naive" CNNs. This would show, for example, how long it takes for the naively initialized networks to catch up to the pre-training; and would give a direct quantification of the advantage of pre-training.
2. The choice of hard sweeps as a proof-of-concept prediction task is a little disappointing, because we can simulate sweeps in a computationally efficient way: simulate the frequency trajectory of a selected mutation, then simulate genealogies from the structured coalescent conditional on that allele frequency trajectory ala Kaplan, Hudson, & Langley (1989). This is the strategy used for simulating selection in discoal (Kern & Schrider 2016) and msprime, and is used to train the CNN-based sweep detector in diploS/HIC (Kern & Schrider 2018). So, it would be good to extend the proof-of-concept with an example that really does preclude existing coalescent simulators: for example, recent adaptive introgression or underdominance or sweeps in the presence of background selection or temporally/geographically-variable selection coefficients. This would help emphasize that the innovation here is a general pre-training strategy, not just another sweep detector.
3. Alternatively, if sweep detection is kept as the only proof-of-concept prediction task, then I think it would be helpful to compare against another method that assumes a misspecified demographic history: for example sweepfinder2, or diploS/HIC trained on a misspecified model, or the CNN architecture in this paper but trained on SLiM simulations with small constant effective population sizes. This would make it clear that there's an advantage to making simulations more realistic. Given the emphasis on the sweep detection results in the paper, the lack of comparison to another method is noticeable.
4. I'm not sure what to take away from the correlations between hidden units and summary statistics (Figures 3D,4D,5D) -- is it just that distinct hidden units pick up on different signals in the data? That the "learned" features recapitulate things we know are important a priori? Either way, it seems like it'd be much more informative to the general reader to just show the classifier score (final logit prediction) vs various summary statistics, under different selection strengths. These would show if the network is learning non-monotonic relationships, if the signal that the neural net is picking up on is "obvious" (e.g. do all the unequivocal scores correspond to regions with extremely low diversity?), etc. The heatmaps in 3D,4D,5D could then be moved to a supplement. Alternatively, please make it clearer why the interpretability analysis is useful.

Small comments:

Ln 18: "are likely to have experienced selection" ==> "are candidates for targets of selection" -- presumably there are many processes in real genomic data that can't be replicated by the generator, of which selection is only one

Ln 60: "identify non-neutral regions" ==> "identify hard selective sweeps" -- the only "non-neutral" model here is hard sweeps, which is just one form of selection

Ln 227: "group neutral regions ... and those with selection strength 0.01 together" -- this is a case where showing classifier scores against summary statistics would be instructive. Is there "human-readable" signal or not? If not, does the neural net still display trends in prediction (vs summary stats), just weaker trends than with larger selection strengths?

Figure 1 caption: "new but similar population" -- reword? "held-out population with similar ancestry"?

Figures 3, 4, 5: Many panels here seem pretty redundant -- I suggest you stick two of them in the supplement and/or combine choice results into a single figure (e.g. color violin plots by population rather than by x-axis value).

Figure 5 / 6: Why are results from YRI/ESN so different from the other two groups of samples? Because there's more polymorphism so there's more power to distinguish false positives? Because there's actually less signal of selection? Because ESN differs from YRI to a greater degree than CHS does from CHB? Some biological or statistical interpretation would be good to include.

Reviewer #2 (Comments for the Authors (Required)):

This is a well-written and interesting manuscript that explores the use of the discriminator half of pg-gan for identifying regions of the genome under positive selection. Rather than following the typical supervised learning procedure of most current methods, this method has parallels to earlier outlier studies, though with a much more sophisticated approach. I think it will be of interest to readers interested in the intersection of cutting-edge machine learning and population genetics.

Comments:

For the architecture described around line 99, can you give a bit more information about where this comes from? I know that the CNN is trained as part of the GAN, but does that apply to every decision (number of kernels, number of layers, size of kernels, etc...?) or does it just apply to all of the weights? If there are things that are not part of the optimization, how are they chosen? If all of these things are part of the optimization, can you give a few sentences about how that works?

line 102: it would be useful to mention here that you train 20 of these for combining later into an ensemble model.

section 2.3: I appreciate that the fine-tuning is meant to help the discriminator pay attention specifically to positive selection, but where does this leave regions of the genome that are neither neutral nor under positive selection? For example, for a region under background selection, what is the expectation for how the discriminator should behave? I could imagine it going either way -- being scored as "real" because it doesn't look neutral, or as "fake" because it doesn't look like positive selection. It would be nice to have an example here of how this method performs on data from some third non-neutral category (maybe in simulation or perhaps across real data; e.g. in exons to examine background selection specifically). If not a new analysis, some discussion would be helpful here.

line 119: "such *that*"

line 125: reference ROC figure panels here

line 134-136 Why does the permutation-invariant feature mean that the number of samples (i.e. rows) can vary without having any issues? I just need a little more hand-holding here.

line 161: I assume the adaptive mutation is added to the center of the simulation? This isn't specified anywhere that I saw.

line 174 bullet points: I was left wondering where the SLIM simulations fit in here. I know they are only used for fine-tuning, but it felt odd to have that part left out. Maybe just some scaffolding here to understand the organization of the study at this point.

line 258: regionregions

line 258-259: I think what you're describing is that the prediction for a single 36-SNP window is the average of the itself, along with 2 downstream and 2 upstream windows. The language was a little unclear though.

line 268-272: I found this hard to follow what was happening and why -- is the permutation just shuffling which regions of the genome you are labeling as under selection? i.e. just to test whether the scores you see across the Grossman set are beyond what you would expect from the same number of randomly placed genomic locations? Or is there something else going on here?

line 272: Nitpicky, but can you rephrase p-value=0.0 to p-value

line 286-287: is it fair to say those stats are implicitly computed when the correlations in Results are not that high?

line 308: "and balancing selection and balancing selection"

Figure 6 caption: "seleciton selection"

General: is there any kind of false discovery analysis that could be done here? Looking at the violin plots in figs 3,4,5 panel C, the distributions look like they have a fair amount of overlap. Does the ensemble approach help here? The analysis in Figure 6 is helpful to this end, but it would be nice to be able to have a better understanding of how many other sites are mixed in with the "ground truth" validated predictions. I recognize this is difficult given how little ground truth we actually have for these populations.

Figs 3,4,5 panels B-C: to what do you attribute the differences between panel B and C? Why does the discriminator have such an easier time on the simulated selection data relative to the real selection data?

Associate Editor Comments:

Two experts in the field have reviewed your manuscript, and I have read it as well. I found the approach and writeup refreshingly creative, clear and forward-thinking. As you will see, both reviewers liked it a good deal as well. While your manuscript is not currently acceptable for publication in GENETICS, we would welcome a substantially revised manuscript. Both reviewers have comments and concerns to be addressed in a revised manuscript. You can read their reviews at the end of this email.

Although the reviewers were very positive about the manuscript, Reviewer 1 (I think correctly) points out that the paper as written is halfway between an exploration of the generic "use a discriminator" approach and a description of a new method for finding hard sweeps. They provide some concrete suggestions for how to add more detail to either of those. (For instance, if this is a "new hard sweep finder" method, then it should probably be compared to an existing one.) I would hope to find some additional depth added to either one of these aspects of the paper in a revised version. Of course, other resolutions (besides those suggested by the reviewer) to this generic concern about a lack of depth are welcome. There is a fair amount of good work in here already, so I would hope that additional experiments are not too onerous. We look forward to receiving your revised manuscript. Please let the editorial office know approximately how long you expect to need for revisions.

We would like to thank the AE and reviewers for the positive and constructive feedback. This way of framing the original manuscript and possibilities for added depth has been very helpful. We agree that the main contributions are the conceptual contributions of 1) using the discriminator for prediction and 2) the fine-tuning approach. We therefore decided to focus our revisions and additional experiments on fine-tuning the trained discriminators on different types of selection, to show how our framework can be thought of as a general purpose transfer learning tool that can be customized for different types of non-neutral regions.

Upon resubmission, please include:

1. A clean version of your manuscript;
2. A marked version of your manuscript in which you highlight significant revisions carried out in response to the major points raised by the editor/reviewers (track changes is acceptable if preferred);
3. A detailed response to the editor's/reviewers' feedback and to the concerns listed above. Please reference line numbers in this response to aid the editor and reviewers.

Your paper will likely be sent back out for review.

Additionally, please ensure that your resubmission is formatted for GENETICS
<https://academic.oup.com/genetics/pages/general-instructions>

Follow this link to submit the revised manuscript:

<https://genetics.msubmit.net/cgi-bin/main.plex?el=A5NR2FfB6A1WLz3l6A9ftd3A61BlxMTamtcl1v6CyUfQZ>

Sincerely,
Peter Ralph

AE comments:

- The parallels to hypothesis testing are amusing, and are probably worth pointing out? (For instance, at least the non-fine-tuned network gives something analogous to a p-value, used as a metric of goodness-of-fit to a particular model.)

Yes this is an interesting parallel. In some sense yes they are like p-values but in another sense, particularly for the fine-tuned discriminators, they are more akin to posterior probabilities - in the sense that we think they actually represent the probability of selection. We have added some discussion about this, pointing out the parallels, as it does provide another lens on how to view outliers (Line 114-115). In general we feel that this question—of how to interpret model outputs in a statistical framework—is an interesting and underappreciated question.

- in the caption for Figure 3, "simulated data with $s=xx$ "... missing something?

$s=xx$ was meant to be a genetic selection coefficient, but we have now changed this to "various selection coefficients" (Fig 3 caption on Page 12)

Additional note: Upon suggestion of the AE, we updated our pyslim version to correct a bug. We also realized that for fine-tuning to work properly with a saved tensorflow neural network, the sample size for the SLiM simulations must match the original training data sample size. We made these modifications and re-ran all the selection simulations, fine-tuning, and downstream analyses (still using the original discriminators trained through pg-gan). Results for individual discriminators changed somewhat, but the ensemble results were remarkably similar, including overlap with known selection signals.

Reviewer #1 (Comments for the Authors (Required)):

This study utilizes adversarial machine learning architectures (GANs), where a parameterized generator (population genetic simulator) is pitted against a discriminator (a neural net classifier) tasked with distinguishing observed data from simulations. Prior work with GANs in population genetics (including by the authors of this paper) has largely focused on the trained generator -- e.g. for the purpose of inferring parameters in some particular evolutionary model or set of models. Here, the authors make the simple and interesting observation that the trained discriminator is useful in its own right, because it has been trained to distinguish features in actual data which are at odds with a "null model" (whatever evolutionary model is employed by the generator) without reference to a particular "alternative model".

Thus, the GAN training regime could be used to pre-train a neural net for tasks that are prohibitively expensive to simulate. So: fit a GAN where the generator is computationally efficient but does not simulate the process of interest, hook the GAN-trained discriminator up to a different loss function/classification task, and fine-tune it with a relatively small number of realistic simulations that *do* include the process of interest. In this case, "prohibitively expensive" means individual-based forward-in-time simulation, whereas "computationally-efficient" means coalescent simulation. The authors suggest that this strategy may make it possible to train neural nets for tasks / demographic settings that would otherwise be computationally infeasible. They apply this strategy to detect hard selective sweeps in human samples, and compare the results to a "truth" set that contains genes that are inferred to be targets of selection a priori (by a previous study). Finally, they correlate values in the last layer of the discriminator (e.g. "features" extracted by the neural net) to summary statistics to provide some insight into what the neural network is learning.

We thank the reviewer for their positive comments on the manuscript.

I agree with the core observation regarding the utility of the discriminator (gave me an 'a-ha!' moment); I think the paper's major point (GANs provide a means to bypass lots of computationally intensive training simulation) is very plausible; and the paper is clearly written despite the complexity of the inference method (there's not too much ML jargon and it should be accessible to many readers). However, in my opinion the most interesting point of the paper (a general-purpose pre-training strategy that can use fast but less realistic simulation) gets overshadowed by the sweep detection proof-of-concept (less interesting, because there's lots and lots of methods to detect sweeps, including ML ones). In particular:

1. The computational savings vs accuracy are never clearly demonstrated. The closest we get are Figs 3A,4A,5A, which compare neural net performance after pre-training and after fine-tuning. However, I don't think this gets the point across. What I'd like to see are training histories for randomly initialized networks as well as for pre-trained discriminators, when fed a series of fine-tuning simulations. Show validation loss against epoch (or alternatively, number of training simulations); I'd think it'd be very clear that the pre-trained CNNs have a huge advantage over the "naive" CNNs. This would show, for example, how long it takes for the naively initialized networks to catch up to the pre-training; and would give a direct quantification of the advantage of pre-training.

This is a great suggestion. We have now fine-tuned randomly initialized networks to assess the impact of GAN pre-training on the discriminators. When considering the training histories we find that the loss starts out much higher for randomly-initialized networks (which is not surprising). As training progresses, the losses eventually approach a similar value (see Fig 6). When considering the accuracy (as measured by ROC curves), we find that often with sufficient fine-tuning, the randomly initialized networks can learn to detect selection in simulation. However, on our validation dataset of real regions from Grossman et al, these networks did not perform as well as the pre-trained+fine-tuned discriminators, often by a wide margin (see Figure 3A compared with Figure 6B). From a classification perspective, this gives us the interesting result that methods that perform similarly in simulation may perform quite differently on real data – being trained with real data throughout the learning process does provide a significant advantage. In that sense the advantage of the pre-training may actually be to improve robustness as much as speed. See section: “Comparison with randomly-initialized networks” for updated manuscript text (Line 301-310).

2. The choice of hard sweeps as a proof-of-concept prediction task is a little disappointing, because we can simulate sweeps in a computationally efficient way: simulate the frequency trajectory of a selected mutation, then simulate genealogies from the structured coalescent conditional on that allele frequency trajectory ala Kaplan, Hudson, & Langley (1989). This is the strategy used for simulating selection in discoal (Kern & Schrider 2016) and msprime, and is used to train the CNN-based sweep detector in diploS/HIC (Kern & Schrider 2018). So, it would be good to extend the proof-of-concept with an example that really does preclude existing coalescent simulators: for example, recent adaptive introgression or underdominance or sweeps in the presence of background selection or temporally/geographically-variable selection coefficients. This would help emphasize that the innovation here is a general pre-training strategy, not just another sweep detector.

Thank you for this excellent suggestion. We have now focused on our method as a general pre-training strategy, with possibilities for transfer learning/fine-tuning using customizable non-neutral simulations. To that end we now provide results for three different types of fine-tuning: positive selection simulations, balancing selection (over-dominance) simulations, and adaptive introgression simulations (Fig 7). Our results suggest that balancing selection is a stronger signal than adaptive introgression (at least for the case of Neanderthal/European introgression when only CEU haplotypes were used). Unfortunately we were not able to validate these on real data; we found that published balancing selection methods had very little overlap, and so we didn't feel that it was viable to use them as validation in the same way that we did for the positive selection case. We found the same thing for Neanderthal introgression scans. We also discuss the possibility of other types of simulations - including the ones you mention as well as recombination hotspots or complex mutational signatures (Line 370-376). Some of these might not be slow to simulation, but we argue there is still value in focusing the majority of training on "neutral" simulations, to mirror the majority of the genome (at least in humans).

3. Alternatively, if sweep detection is kept as the only proof-of-concept prediction task, then I think it would be helpful to compare against another method that assumes a misspecified demographic history: for example sweepfinder2, or diploS/HIC trained on a misspecified model, or the CNN architecture in this paper but trained on SLiM simulations with small constant effective population sizes. This would make it clear that there's an advantage to making simulations more realistic. Given the emphasis on the sweep detection results in the paper, the lack of comparison to another method is noticeable.

This is fair and we have decided to focus on (2), emphasizing that this is a general purpose strategy (i.e. training a GAN to detect real vs. simulated and then fine-tuning the discriminator with custom simulations) rather than claim this is a superior sweep detection method.

4. I'm not sure what to take away from the correlations between hidden units and summary statistics (Figures 3D,4D,5D) -- is it just that distinct hidden units pick up on different signals in the data? That the "learned" features recapitulate things we know are important a priori? Either way, it seems like it'd be much more informative to the general reader to just show the classifier score (final logit prediction) vs various summary statistics, under different selection strengths. These would show if the network is learning non-monotonic relationships, if the signal that the neural net is picking up on is "obvious" (e.g. do all the unequivocal scores correspond to regions with extremely low

diversity?), etc. The heatmaps in 3D,4D,5D could then be moved to a supplement. Alternatively, please make it clearer why the interpretability analysis is useful.

This is a great suggestion - we did not initially include this analysis since the one output from the discriminator likely does not capture all of the summary statistics we computed (which was the motivation for analyzing the hidden units, to see if they were “specializing” in different summary statistics). We have now included representative plots (Fig S5) for various selection strengths. For neutral data and low selection coefficients, we see similar patterns where π is very important. As the selection coefficient increases, we see that both the number of singletons and π are very important, and then for LD, the LD values for closer SNPs are negatively correlated with the prediction, and SNPs that are further away from each other, their LD values are positively correlated with the prediction. Overall we believe interpretability analysis should be standard for all population genetic ML, to emphasize that ML methods are connected to our biological intuition.

Small comments:

Ln 18: "are likely to have experienced selection" ==> "are candidates for targets of selection" -- presumably there are many processes in real genomic data that can't be replicated by the generator, of which selection is only one

We have now modified this part to be more general: “regions with a high probability of being real do not fit the neutral demographic model”

Ln 60: "identify non-neutral regions" ==> "identify hard selective sweeps" -- the only "non-neutral" model here is hard sweeps, which is just one form of selection

We have retained this text, since we now include three forms of non-neutrality. Conversely, things like biased gene conversion are neutral processes indistinguishable from selection.

Ln 227: "group neutral regions ... and those with selection strength 0.01 together" -- this is a case where showing classifier scores against summary statistics would be instructive. Is there "human-readable" signal or not? If not, does the neural net still display trends in prediction (vs summary stats), just weaker trends than with larger selection strengths?

This is a great point, and we have now included these plots (classifier scores against summary statistics) in the Supplement (Fig S5). We do see a strong trend as the

selection coefficient increases, although for weak selection we still don't see much separation between neutral and these low selection coefficients. We suspect that in practice it's just not really possible to infer such weak selection without time series data - few published methods are able to identify $s < 0.01$, even with large sample sizes.

Figure 1 caption: "new but similar population" -- reword? "held-out population with similar ancestry"?

We have now changed this to the suggested wording (Fig 1 caption), which is much more accurate.

Figures 3, 4, 5: Many panels here seem pretty redundant -- I suggest you stick two of them in the supplement and/or combine choice results into a single figure (e.g. color violin plots by population rather than by x-axis value).

We have now retained the CEU figure and put CHB and YRI in the supplement (Figs S3 and S4).

Figure 5 / 6: Why are results from YRI/ESN so different from the other two groups of samples? Because there's more polymorphism so there's more power to distinguish false positives? Because there's actually less signal of selection? Because ESN differs from YRI to a greater degree than CHS does from CHB? Some biological or statistical interpretation would be good to include.

Overall we found that the results on simulated selection data (in terms of ROC curves) were somewhat better for YRI, but for the real validation data, the YRI/ESN ROC curve is not as accurate. It is definitely true that the YRI/ESN difference is greater than CHB/CHS and that could be driving some of the poor performance. However, we hypothesize this is more to do with the Grossman et al selection results being potentially incomplete/inaccurate for YRI more so than for CEU and CHB. We've added the following text (Line 272-275):

"In simulated data, predictions are better for YRI demography, compared to CEU or CHB, but for real data this pattern is reversed. We suspect that power is theoretically higher for YRI due to higher diversity, but that the selection signals from \cite{grossman2013identifying} are either less reliable or less shared between YRI/ESN (compared to CEU/GBR or CHB/CHS)."

Reviewer #2 (Comments for the Authors (Required)):

This is a well-written and interesting manuscript that explores the use of the discriminator half of pg-gan for identifying regions of the genome under positive selection. Rather than following the typical supervised learning procedure of most current methods, this method has parallels to earlier outlier studies, though with a much more sophisticated approach. I think it will be of interest to readers interested in the intersection of cutting-edge machine learning and population genetics.

We thank the reviewer for their positive comments on the manuscript.

Comments:

For the architecture described around line 99, can you give a bit more information about where this comes from? I know that the CNN is trained as part of the GAN, but does that apply to every decision (number of kernels, number of layers, size of kernels, etc...?) or does it just apply to all of the weights? If there are things that are not part of the optimization, how are they chosen? If all of these things are part of the optimization, can you give a few sentences about how that works?

This is a great point and we have now included additional information about both the CNN architecture and how it is optimized (Line 100-107). During pg-gan training only the weights are optimized, but it could be a very interesting future direction to also optimize parts of the architecture during training as well (i.e. number of layers, kernel sizes, etc). This could be seen as a way of incorporating hyper-parameter tuning into GAN training. I don't think this is common in other GANs for technical reasons (i.e. how to initialize new weights while some are partially trained) but it could be possible and fruitful for creating the best match between simulated and real data.

line 102: it would be useful to mention here that you train 20 of these for combining later into an ensemble model.

We have now included this information (Line 107-108).

section 2.3: I appreciate that the fine-tuning is meant to help the discriminator pay attention specifically to positive selection, but where does this leave regions of the genome that are neither neutral nor under positive selection? For example, for a region under background selection, what is the expectation for how the discriminator should behave? I could imagine it going either way -- being scored as "real" because it doesn't look neutral, or as "fake" because it doesn't look like positive selection. It would be nice

to have an example here of how this method performs on data from some third non-neutral category (maybe in simulation or perhaps across real data; e.g. in exons to examine background selection specifically). If not a new analysis, some discussion would be helpful here.

Related to your point about false discovery (see below), this is a good point and we now include an expanded discussion about other forces that could be causing high (or low) discriminator predictions (Line 359-369). One concrete way to investigate this might be to look at summary statistics for regions where many of our discriminators had a high prediction, but the region was not in or near a known gene. Summary statistics could help us entangle if these regions have some other notable feature such as high LD, excess of singletons, low/high SNP density, etc.

line 119: "such *that*"

Fixed.

line 125: reference ROC figure panels here

Done (Line 131).

line 134-136 Why does the permutation-invariant feature mean that the number of samples (i.e. rows) can vary without having any issues? I just need a little more hand-holding here.

This definitely warrants a bit more detail, which we have now included (Line 142-146). Typically in a non-permutation-invariant CNN, at some point the output of the convolutional layers has to be unraveled into a fully-connected layer, which constrains the shape of train/test inputs to be the same. With our custom permutation-invariant CNN, after the convolutional layers we collapse the information across haplotypes to a single value. So if we have 196 vs. 200 haplotypes, the architecture will still work without dimension mismatch runtime errors.

line 161: I assume the adaptive mutation is added to the center of the simulation? This isn't specified anywhere that I saw.

Yes that's correct and we have now included this information (Line 171).

line 174 bullet points: I was left wondering where the SLiM simulations fit in here. I know they are only used for fine-tuning, but it felt odd to have that part left out. Maybe just some scaffolding here to understand the organization of the study at this point.

That's fair and we have now included the SLiM simulations in this section (Line 197). It is now about validation in general (using both positive selection simulations and known selected regions).

line 258: regionregions

Fixed.

line 258-259: I think what you're describing is that the prediction for a single 36-SNP window is the average of the itself, along with 2 downstream and 2 upstream windows. The language was a little unclear though.

Yes exactly right - we have now used this language to make the smoothing description clearer (Line 283-285).

line 268-272: I found this hard to follow what was happening and why -- is the permutation just shuffling which regions of the genome you are labeling as under selection? i.e. just to test whether the scores you see across the Grossman set are beyond what you would expect from the same number of randomly placed genomic locations? Or is there something else going on here?

Yes that's exactly right - we have now clarified this section to make it clear we are just shuffling which regions of the genome are labeled as under selection (while keeping the original region lengths the same). We calculate the average of our predictions inside and outside these regions to obtain two numbers (for each shuffling) for comparison (Line 292-297).

line 272: Nitpicky, but can you rephrase $p\text{-value}=0.0$ to $p\text{-value}<\text{threshold}$ where threshold is whatever your minimum possible value would be?

We have now rephrased this (basically all p-values rounded to 0.0 in python, so we have included the threshold where that happens) (Line 298).

line 286-287: is it fair to say those stats are implicitly computed when the correlations in Results are not that high?

That's fair - we have now rephrased this part (Line 322). We also note that for the final discriminator prediction (for selected regions) we see higher correlations (Fig S5).

line 308: "and balancing selection and balancing selection"

No longer included in the text.

Figure 6 caption: "seleciton selection"

Fixed.

General: is there any kind of false discovery analysis that could be done here? Looking at the violin plots in figs 3,4,5 panel C, the distributions look like they have a fair amount of overlap. Does the ensemble approach help here? The analysis in Figure 6 is helpful to this end, but it would be nice to be able to have a better understanding of how many other sites are mixed in with the "ground truth" validated predictions. I recognize this is difficult given how little ground truth we actually have for these populations.

This is a good point and we now include an expanded discussion on the topic of false discovery (Line 359-369). That said, as you suggest it's just hard to get a qualitative sense of this since the False Negative rate for the other selection scans we're using as validation is almost entirely unknown. For regions where many discriminators output a high probability, we also used the human genome browser to see if there were any genomic features of interest in these regions. In some cases (included in the excel spreadsheet in the Supplement) our regions did overlap with other genes (not identified by Grossman *et al*). However, in other cases it is unclear what is going on.

Figs 3,4,5 panels B-C: to what do you attribute the differences between panel B and C? Why does the discriminator have such an easier time on the simulated selection data relative to the real selection data?

We attribute this to the training procedure (specifically fine-tuning), where the discriminator was shown simulated regions where we are 100% confident there was selection. When shown similar regions during testing, it is typically easily able to identify these regions as selected. In contrast, the real selected regions were identified by another method, and may include both false negatives and false positives. Further, the real data include other biological processes (for example heterogeneity in mutation, recombination rate or other types of selection) that are not reflected in the simulated data.

December 12, 2023
RE: GENETICS-2023-306637

Dear Dr. Mathieson:

I am pleased to accept your manuscript entitled "Interpreting Generative Adversarial Networks to Infer Natural Selection from Genetic Data" for publication in GENETICS, pending minor revision.

As you'll see, both reviewers have a number of additional suggestions, but most of these entail some minor adjustments to the wording. Both reviewers do suggest additional analyses, but please regard these as strictly optional (as also suggested by the reviewers). Please submit your revision along with a brief description of how you modified the manuscript in response to the reviewers' concerns and suggestions (which can be viewed at the bottom of this email. (Point-by-point responses are not required.) I expect you should be able to submit a revised manuscript within 30 days or so (accounting for holidays). A suitably revised manuscript will be acceptable for publication; I don't expect to send it out for review.

When revising the ms., please make an effort to shorten it, because that almost always improves a manuscript. We urge authors to heed the advice of Strunk and White: "omit needless words"¹. Follow this link to submit the revised manuscript: Link Not Available

Thank you for submitting this story to Genetics.

Sincerely,

Peter Ralph
Associate Editor
GENETICS

Approved by:
Nicholas Barton
Senior Editor
GENETICS

Reviewer comments:

Reviewer #1 (Comments for the Authors (Required)):

The authors have addressed my criticisms with the original manuscript. I have one substantial suggestion, which is to further emphasize the result that the GAN-trained CNN does much better than the randomly initialized CNN on real data, even after the latter has been trained to a similar validation loss. In my opinion, this seems just as important as any computational savings, because it reflects the ability of the GAN to adapt to unmodelled processes in actual data, and to transfer these to subsequent prediction tasks. This seems like a really neat aspect and is definitely worth bringing up again in the discussion.

Ln 129: "new genetic regions" ==> "new" in the sense that the predictions should be performed on data that the GAN wasn't trained on (the 'held-out' population mentioned in figure 1), I'm assuming? As this is the first time that the hold-out comes up, it'd be good to be explicit.

Ln 147-149: Are point estimates for these demographic parameters used, or are you drawing random values from an approximate posterior?

Ln 209: "significant" in the last line of pseudocode is wrt the posterior probability of the region being neutral? Where the averaged discriminator output approximates the posterior probability?

Ln 308-309: the discrepancy on real data makes sense, and is a really neat result -- I suggest bringing it up again in the discussion! To me, the ability to adapt to unmodelled processes seems just as important as computational savings, if not more so.

Fig 7: I was hoping to see results for a randomly initialized CNN here (after, say, 100 batches of training on selection simulations), akin to what is shown in Fig 6. But, I recognize that this paper already contains a lot of material (more than enough to illustrate the utility of the method).

Ln 335-336: So, given that the randomly initialized CNN in Fig 7 takes ~20 batches to reach the starting loss of the pre-trained CNN, are there any computational savings from pre-training? That is, are the 5-7 hours for msprime simulation + pg-gan training less than the time needed for 20 batches of SLiM simulations?

I bring this up because it's highlighted in the discussion (Ln 347-351) but there's not a clear quantification (e.g. a back-of-the-envelope calculation, based on batch size used for randomly initialized CNN and assuming simulations are generated on the fly).

Ln 359: Couldn't you use the pretrained discriminator, but hook up a regression output to predict the selection coefficient, rather than a classification probability? That is, the pretrained discriminator has presumably learned to engineer features, and this is likely useful even if the weights in the finally fully connected layers need more fine-tuning to adapt to a continuous output.

Figure S5: Looks great. The bimodal pattern for the iter-SNP distances that emerges for large selection coefficients is interesting.

Reviewer #2 (Comments for the Authors (Required)):

I think the manuscript is really strong and is better for the changes made in the first round of revisions. The narrative is much clearer, and it benefits from the additional exploration of other modes of selection. I have some minor comments below:

1. As I was reading, my brain kept getting stuck on something until I had a sort of epiphany halfway through that I think is worth stating up front (if it is correct). With the GAN architecture, since the generator is learning to produce data that is as close to real data as possible, I couldn't figure out why it wouldn't just learn to make data that looks like all parts of the genome (neutral and not), in which case the designation of "fake" and "real" to refer to "neutral" and "selection" seemed less clear-cut, and it would seem that the generator should keep improving to the point where the discriminator can no longer tell the difference. What I think I realized is that because of pg-gan's demographic model-based approach, it explicitly can only model neutral processes, so it is forced to do its best to approximate real data without being able to actually include selection as a demographic event. This strikes me as a real benefit of this model-based approach that pg-gan uses -- a generator that is not restricted in this way could potentially learn to simulate data that spans the gamut of neutral and non-neutral genome regions (in some sense, doing a better job at mimicking real data), which would make what you're doing in this paper a lot harder. If this interpretation is correct, I think it would be helpful to have something in the introduction to this effect.

I noticed that there is a sentence that says something like this in the abstract (line 16), but I was confused by its wording when I first came across it (including the emphasis on real "training" data rather than real data more generally, in either training or testing) and the concept didn't sink in. Perhaps the abstract sentence could be something like: "As the generator only has access to neutral demographic processes, regions in real data that the discriminator recognizes as having a high probability of being 'real' do not fit the neutral demographic model and are therefore likely to be non-neutral and possibly undergoing selection"

2. Point #1 did bring up a followup question for me though: since the generator is being optimized to produce data as similar to real data as possible, but it is constrained to only neutral processes, is there any danger of misspecification? For example, if you had a population subject to a lot of positive selection, would the generator substitute this with historical bottlenecks that didn't actually exist? I think you have evidence in this paper that this is not happening in your case (in Figure 2 e.g.) but some discussion would be nice here.

3. Line 19: should this be "with a small number of selection simulations of a particular type" (i.e. positive, balancing, adaptive introgression)?

4. Line 47: "make""made"

5. Line 209 (Algorithm 1): "if the result is significant..." how is significance determined here?

6. Line 213-215: This wording was confusing. Maybe something like "...we computed the correlation between various summary statistics and parts of our discriminator network: the node values of the last hidden layer and the final prediction."

7. Line 219: "networks""network's"

8. I was left wanting a little bit more with the new fine-tuned discriminators for balancing selection and adaptive introgression, and at points throughout I was a little confused whether the results/methods being discussed referred to positive selection only,

or all 3 modes of selection. For example, I think the genome results and interpretability analysis only apply to positive selection, but it would have been nice to see these for the other two (or even just for balancing selection, since the validation results look pretty good for that). These additional analyses aren't deal-breakers; they would just make for a more complete story.

Associate Editor comments:

I am pleased to accept your manuscript entitled "Interpreting Generative Adversarial Networks to Infer Natural Selection from Genetic Data" for publication in GENETICS, pending minor revision.

As you'll see, both reviewers have a number of additional suggestions, but most of these entail some minor adjustments to the wording. Both reviewers do suggest additional analyses, but please regard these as strictly optional (as also suggested by the reviewers). Please submit your revision along with a brief description of how you modified the manuscript in response to the reviewers' concerns and suggestions (which can be viewed at the bottom of this email. (Point-by-point responses are not required.) I expect you should be able to submit a revised manuscript within 30 days or so (accounting for holidays). A suitably revised manuscript will be acceptable for publication; I don't expect to send it out for review.

When revising the ms., please make an effort to shorten it, because that almost always improves a manuscript. We urge authors to heed the advice of Strunk and White: "omit needless words"¹. Follow this link to submit the revised manuscript:

<https://genetics.msubmit.net/cgi-bin/main.plex?el=A3NR5Fsu2A1WLz5I4A9ftdkbv4i69YESmybi4XhAznAZ>

Thank you for submitting this story to Genetics.

Sincerely,

Peter Ralph
Associate Editor
GENETICS

Approved by:
Nicholas Barton
Senior Editor
GENETICS

Reviewer comments:

Reviewer #1 (Comments for the Authors (Required)):

The authors have addressed my criticisms with the original manuscript. I have one substantial suggestion, which is to further emphasize the result that the GAN-trained CNN does much better than the randomly initialized CNN on real data, even after the latter has been trained to a similar validation loss. In my opinion, this seems just as important as any computational savings, because it reflects the ability of the GAN to adapt to unmodelled processes in actual data, and

to transfer these to subsequent prediction tasks. This seems like a really neat aspect and is definitely worth bringing up again in the discussion.

We agree and were very intrigued to see this result. In population genetics we rarely have access to this type of experiment, but here our real validation dataset (i.e. Grossman et al selected regions) gives us a sense of how well our method is doing on real data. It suggests that many methods might perform similarly well on simulations, but show very different results for real data. We have now emphasized this further in the discussion.

Ln 129: "new genetic regions" ==> "new" in the sense that the predictions should be performed on data that the GAN wasn't trained on (the 'held-out' population mentioned in figure 1), I'm assuming? As this is the first time that the hold-out comes up, it'd be good to be explicit.

We have now made this more explicit.

Ln 147-149: Are point estimates for these demographic parameters used, or are you drawing random values from an approximate posterior?

This part is just to explain the general parameter fitting process (which produces point estimates). When we create non-neutral simulations we use these point estimates (not an approximate posterior), which we've highlighted in the caption for Table 2.

Ln 209: "significant" in the last line of pseudocode is wrt the posterior probability of the region being neutral? Where the averaged discriminator output approximates the posterior probability?

Thanks for catching this - in our previous revisions we dropped the notion of genome-wide significance and interpreted the results directly as probabilities (akin to approximate posterior probabilities of selection). We have rephrased this part of the algorithm.

Ln 308-309: the discrepancy on real data makes sense, and is a really neat result -- I suggest bringing it up again in the discussion! To me, the ability to adapt to unmodelled processes seems just as important as computational savings, if not more so.

Agreed - we have now reemphasized this in the discussion.

Fig 7: I was hoping to see results for a randomly initialized CNN here (after, say, 100 batches of training on selection simulations), akin to what is shown in Fig 6. But, I recognize that this paper already contains a lot of material (more than enough to illustrate the utility of the method).

This is a great idea, but given the lack of real validation dataset for these cases (balancing selection and adaptive introgression), I'm not sure we would see the benefit of pre-training as clearly. I'm guessing we would see a similar pattern of randomly-initialized networks starting out with a high loss but eventually doing fairly well in simulations.

Ln 335-336: So, given that the randomly initialized CNN in Fig 7 takes ~20 batches to reach the starting loss of the pre-trained CNN, are there any computational savings from pre-training? That is, are the 5-7 hours for msprime simulation + pg-gan training less than the time needed for 20 batches of SLiM simulations?

I bring this up because it's highlighted in the discussion (Ln 347-351) but there's not a clear quantification (e.g. a back-of-the-envelope calculation, based on batch size used for randomly initialized CNN and assuming simulations are generated on the fly).

That's a great question. With 50 regions per batch, 20 batches would be 1000 SLiM simulations. The timing for SLiM simulations depends on the demography and the selection strength, but was 4-8 hours per 100 simulations for CEU. These can be parallelized, but if they are not, then I would say there are some computational savings to pre-training. We have now summarized this in the Runtime section.

Ln 359: Couldn't you use the pretrained discriminator, but hook up a regression output to predict the selection coefficient, rather than a classification probability? That is, the pretrained discriminator has presumably learned to engineer features, and this is likely useful even if the weights in the finally fully connected layers need more fine-tuning to adapt to a continuous output.

That's a great idea and we have now mentioned it as a possibility in the discussion.

Figure S5: Looks great. The bimodal pattern for the iter-SNP distances that emerges for large selection coefficients is interesting.

Thanks!

Reviewer #2 (Comments for the Authors (Required)):

I think the manuscript is really strong and is better for the changes made in the first round of revisions. The narrative is much clearer, and it benefits from the additional exploration of other modes of selection. I have some minor comments below:

1. As I was reading, my brain kept getting stuck on something until I had a sort of epiphany halfway through that I think is worth stating up front (if it is correct). With the GAN architecture, since the generator is learning to produce data that is as close to real data as possible, I couldn't figure out why it wouldn't just learn to make data that looks like all parts of the genome (neutral and not), in which case the designation of "fake" and "real" to refer to "neutral" and "selection" seemed less clear-cut, and it would seem that the generator should keep improving to the point where the discriminator can no longer tell the difference. What I think I realized is that because of pg-gan's demographic model-based approach, it explicitly can only model neutral processes, so it is forced to do its best to approximate real data without being able to actually include selection as a demographic event. This strikes me as a real benefit of this

model-based approach that pg-gan uses -- a generator that is not restricted in this way could potentially learn to simulate data that spans the gamut of neutral and non-neutral genome regions (in some sense, doing a better job at mimicking real data), which would make what you're doing in this paper a lot harder. If this interpretation is correct, I think it would be helpful to have something in the introduction to this effect.

I noticed that there is a sentence that says something like this in the abstract (line 16), but I was confused by its wording when I first came across it (including the emphasis on real *training* data rather than real data more generally, in either training or testing) and the concept didn't sink in. Perhaps the abstract sentence could be something like: "As the generator only has access to neutral demographic processes, regions in real data that the discriminator recognizes as having a high probability of being 'real' do not fit the neutral demographic model and are therefore likely to be non-neutral and possibly undergoing selection"

This is a great point, and one that we have now emphasized more clearly (including a version of your wording in the abstract, which is very helpful). In fact, restricting the pre-training to a predefined set of evolutionary constraints (neutrality, etc) could allow this approach to be even more general.

2. Point #1 did bring up a followup question for me though: since the generator is being optimized to produce data as similar to real data as possible, but it is constrained to only neutral processes, is there any danger of misspecification? For example, if you had a population subject to a lot of positive selection, would the generator substitute this with historical bottlenecks that didn't actually exist? I think you have evidence in this paper that this is not happening in your case (in Figure 2 e.g.) but some discussion would be nice here.

I believe this is a caveat of our approach and we have now added this language to the discussion. For humans, strong selection is relatively rare across the genome, but for other species (i.e. drosophila) this is not the case. If a large fraction of the genome is under selection, I'm guessing this would warp the (neutral) demographic inference somehow (as it would for other methods), but it's possible it would simply lead to more regions with high probabilities of being "real". That said, this is a problem for all methods of detecting selection and our approach is conservative in the sense that it explains as much of the variation as it can using neutral processes.

3. Line 19: should this be "with a small number of selection simulations of a particular type" (i.e. positive, balancing, adaptive introgression)?

Thanks for catching this - we have now modified the language to make it more general.

4. Line 47: "make""made"

Fixed.

5. Line 209 (Algorithm 1): "if the result is significant..." how is significance determined here?

Thanks for catching this - in our previous revisions we dropped the notion of genome-wide significance and interpreted the results directly as probabilities (akin to approximate posterior probabilities of selection). We have rephrased this part of the algorithm.

6. Line 213-215: This wording was confusing. Maybe something like "...we computed the correlation between various summary statistics and parts of our discriminator network: the node values of the last hidden layer and the final prediction."

Thanks - this wording is much better.

7. Line 219: "networks""network's"

Fixed.

8. I was left wanting a little bit more with the new fine-tuned discriminators for balancing selection and adaptive introgression, and at points throughout I was a little confused whether the results/methods being discussed referred to positive selection only, or all 3 modes of selection. For example, I think the genome results and interpretability analysis only apply to positive selection, but it would have been nice to see these for the other two (or even just for balancing selection, since the validation results look pretty good for that). These additional analyses aren't deal-breakers; they would just make for a more complete story.

That's fair and we have now tried to make it more clear when an analysis was performed only for the positive selection case. We felt the genome-wide analysis was most informative in the positive selection case where well-known signals have been established, but we also ran interpretability analysis for the balancing selection case (since the validation results were quite good as you mention). However, there were not as clear correlations between the network predictions and summary statistics. We hypothesize statistics that measure balancing selection are more complex, and sensitive to the particular mode of selection, which is why we don't see strong correlations with this set of relatively standard summary statistics. We have included a summary of this analysis in the main text.

January 19, 2024

RE: GENETICS-2023-306637R1

Dr. Sara Mathieson
Haverford College
Computer Science
370 Lancaster Ave
Haverford

Dear Dr. Mathieson:

Congratulations! We are delighted to inform you that your manuscript entitled "Interpreting Generative Adversarial Networks to Infer Natural Selection from Genetic Data" is acceptable for publication in GENETICS. Many thanks for submitting your research to the journal.

I have read the revised version carefully, and (found it very clear and insightful and) have a few comments you may want to take into consideration for the final version (below).

To Proceed to Production:

1. Format your article according to GENETICS style, as discussed at <https://academic.oup.com/genetics/pages/general-instructions>, and upload your final files at <https://genetics.msubmit.net>.
2. Your manuscript will be published as-is (unedited-as submitted, reviewed, and accepted) at the GENETICS website as an Advanced Access article and deposited into PubMed shortly after receipt of source files and the completed license to publish. Please notify sourcefiles@thegsajournals.org if you do not wish to publish your article via Advanced Access.
3. We invite you to submit an original color figure related to your paper for consideration as cover art. Please email your submission to the editorial office or upload it with your final files. You can submit a small-sized image for evaluation, and if selected, the final image must be a TIFF file 2513px wide by 3263px high (8.375 by 10.875 inches; resolution of 600ppi). Please avoid graphs and small type.

If you have any questions or encounter any problems while uploading your accepted manuscript files, please email the editorial office at sourcefiles@thegsajournals.org.

Sincerely,

Peter Ralph
Associate Editor
GENETICS

Approved by:
Nicholas Barton
Senior Editor
GENETICS

note: Please add jnls.author.support@oup.com and genetics.oup@kwglobal.com (or the domains @oup.com and @kwglobal.com) to your email program's "safe senders" list. You will be contacted by both at various points during the production process.

AE comments:

The paper refers to the output as a "probability of selection"; however, it is not a well-calibrated probability in any sense (unless I've missed something?). I'm not particularly bothered by this use of the word "probability", but I do think that a fairly strong warning to readers about overinterpretation should be issued at some point. (For instance, it's probably not true that 20% of

regions whose 'probability of selection' is 0.2 are actually under selection;
but if you disagree, explain?)

p.4 l.104: "The convolutional layers have 32 and 64 filters (respectively) the fully connected layers": missing conjunction?

Table 2 caption: "There point estimates" -> "These..."

Figure 3 caption: The labels (B,C,D) are mixed up (cyclically permuted?).